

# Interpreting the time variability of world-wide GPS and GOME/SCIAMACHY integrated water vapour retrievals, using reanalyses as auxiliary tools

Roeland Van Malderen[1], Eric Pottiaux[2], Gintautas Stankunavicius[3], Steffen Beirle[4], Thomas Wagner[4], Hugues Brenot[5], and Carine Bruyninx[2]

[1]Royal Meteorological Institute of Belgium (RMIB), Uccle, Belgium
[2]Royal Observatory of Belgium (ROB), Uccle, Belgium
[3]Dep. of Hydrology and Climatology, Institute of Geosciences, Faculty of Chemistry and Geosciences, Vilnius University, Lithuania
[4]Max Planck Institute for Chemistry (MPI-C), Mainz, Germany
[5]Royal Belgium Institute for Space Aeronomy (BIRA), Uccle, Belgium

*Correspondence to*: Roeland Van Malderen (Roeland.VanMalderen@meteo.be)

**Abstract.** This study investigates different aspects of the Integrated Water Vapour (IWV) variability at 118 globally distributed Global Positioning System (GPS) sites, using additionally UV/VIS satellite retrievals by GOME, SCIAMACHY and GOME-2 (denoted as GOMESCIA below), and ERA-Interim reanalysis output at these site locations. Apart from some spatial representativeness issues at especially coastal and island sites, those three datasets correlate rather well, the lowest correlation found between GPS and GOMESCIA (0.865 on average). In this paper, we first study the geographical distribution of the frequency distributions of the IWV time series, and subsequently analyse the seasonal IWV cycle and linear trend differences among the three different datasets. Finally, both the seasonal behaviour and the long-term variability are fitted together by means of a stepwise multiple linear regression of the station's time series, with a selection of regionally dependent candidate explanatory variables. Overall, the variables that are most frequently used and explain the largest fractions of the IWV variability are the surface temperature and precipitation. Also the surface pressure and tropopause pressure (in particular for higher latitude sites) are important contributors to the IWV time variability. All these variables also seem to account for the sign of long-term trend in the IWV time series to a large extent, when considered as explanatory variable. Furthermore, the multiple linear regression linked the IWV variability at some particular regions to teleconnection patterns or climate/oceanic indices like the North Oscillation index for West USA, the El Niňo Southern Oscillation (ENSO) for East Asia, the East Atlantic (associated with the North Atlantic Oscillation, NAO) index for Europe.

## 1 Introduction

Being the most important natural greenhouse gas and responsible for the largest known feedback mechanism for amplifying climate change (Soden and Held, 2006), the role of water vapour is crucial in a warming climate. As a matter of fact, the



amount and the time variability of atmospheric water vapour is globally governed by the temperature through the Clausius-Clapeyron equation which states that the water holding capacity increases at about 7% per degree Celsius increment in temperature (Trenberth et al., 2005; Wentz et al., 2007). This increase rate (5-7%) was confirmed over tropical oceans by Mears et al. (2007), using both climate models and satellite observations. Of course, on local scales, water vapour also

strongly influences atmospheric dynamics and the hydrologic cycle through latent heat transport and diabatic heating, and is, in particular, a source of clouds and precipitation.

Atmospheric water vapour is highly variable, both in space and in time. Therefore, measuring it remains a demanding and challenging task. Numerous techniques measuring the Integrated Water Vapour (IWV) amount exist, from ground-based devices, in-situ (radiosondes) and from space on board satellites. Guerova et al. (2016) divided remote sensing techniques

into 3 categories: (1) differential time of arrival measurements, like Global Navigation Satellite Systems (GNSS), such as GPS, Very Long Baseline Interferometry (VLBI), and Doppler Orbitography Radio positioning Integrated by Satellite (DORIS), (2) active techniques like LIght Detection And Ranging (LIDAR) and RAdio Detection And Ranging (RADAR), (3) passive techniques based on emission/absorption measurements.. Among ground-based emission/absorption measurements there are microwave radiometers (e.g. Morland et al., 2009, Hocke et al., 2011), photometers (with the sun,

the moon or a star as light source, see e.g. Pérez-Ramírez et al., 2014, Campanelli et al., 2018),  Fourier-Transform Infrared Spectroscopy (FTIR, see e.g. Schneider et al., 2010, Buehler et al., 2012), and multi-axis differntial optical absorption spectroscopy (MAX-DOAS, see e.g. Irie et al., 2011, Wagner et al., 2013). Also space measurements of IWV make advantage of different parts of the electromagnetic spectrum of the Earth's atmosphere: microwave (AMSU-B, AMSR-E, SSM/I, SSMIS, HSB, etc.), visible (OMI, GOME, GOME-2, SCIAMACHY, etc.), near-infrared (MODIS) and thermal

infrared (MODIS, AIRS, SEVIRI, IASI, ISCCP (TOVS), etc.). An inventory many of those available satellite data records has been carried out within the Global Energy and Water Exchanges (GEWEX) Water Vapour Assessment (G-VAP) and made available online at http://gewex-vap.org/?page_id=309. All the mentioned different techniques have their proper advantages and disadvantages, and their IWV retrievals have been inter-compared in numerous studies (see e.g. http://www.meteo.be/IWVintercomp for a literature overview of past IWV inter-technique studies involving GNSS). In such

a study (Van Malderen et al., 2014), we compared five different techniques and concluded that the observations used here (GPS and GOMESCIA) have the potential to be used for climate change analysis, which is the subject of this paper.

The time variability of IWV has already extensively been studied in the literature, based on several observational datasets (radiosondes, GNSS, DORIS, microwave and visible satellite measurements) as well as reanalyses. Globally, the mean IWV distribution follows the mean temperature distribution and is highest at the tropics, where strong evaporation occurs and

trade winds transport the moisture to the Intertropical Convergence Zone (Parracho et al., 2018). At mid and high latitudes, the evaporation is weaker and lower IWV amounts are observed. Furthermore, the highest mean IWV values occur in the warm (summer) season, the lowest values in the cold (winter) months. Global decadal IWV trend studies (partly summarised until 2009 in Sherwood et al. (2010), and thereafter in Bock et al. (2014), Mieruch et al. (2014), Chen and Liu (2016), Schröder et al. (2016), Wang et al. (2016a), Mears et al., (2018), and Parracho et al. (2018)) reveal a positive overall global



mean trend, e.g. of 0.26, 0.24, and 0.34 mm dec$^{-1}$, respectively, in the GPS (1995-2011), radiosonde (1973-2011), and microwave satellite (1988-2011) records (Wang et al., 2016a). But the significance and the sign of the trends can change from region to region, from dataset to dataset, and from study to study, as pointed out by e.g. Wagner et al. (2006), Mieruch et al. (2014), and Schröder et al. (2016). Of course, these trends can be influenced by several factors, like the spatial

coverage and time span considered, the trend estimation method, the homogeneity of the datasets, but also natural phenomena like autocorrelation and ENSO (see e.g. Wagner et al., 2006, Shi et al., 2018, Wang et al., 2018).

In this paper, we study the IWV time variability by means of a stepwise multiple linear regression approach for a worldwide sample of GPS sites, at which satellite IWV measurements and reanalysis IWV output are also considered. To our knowledge, it is the first time that such an analysis is done on the IWV time series of individual sites. But first, we describe

(in Sect. 2) and compare (in Sect. 3) the different IWV datasets used in this study. In Sect. 4, we analyse the geographical distribution of the IWV frequency distributions of the stations. The seasonal behaviour is studied in Sect. 5, while the linear trends are presented in Sect. 6.1. The stepwise multiple linear regression analysis itself can be found in Sect. 6.2. The last Sect. 7 is reserved for conclusions.

## 2 Datasets

For the determination of the IWV time variability over land, we make use of two different datasets: IWV retrievals from permanently observing GPS stations at one hand, and a merged dataset of IWV measured with 3 different satellite instruments (GOME, SCIAMACHY, GOME-2) at the other hand. In addition, we use the IWV from reanalyses, ERA-Interim and, to a lesser extent, NCEP/NCAR, as auxiliary tools to check the consistency of the features revealed by the observational IWV datasets.

### 2.1 GPS

Time delay measurements in the signal propagation between satellites and a network of ground-based GPS receivers are partly due to the amount of water vapour in the neutral atmosphere (and vastly in the troposphere). By removing from measurements of the (total) delay induced by the neutral atmosphere (i.e. the Zenith Total Delay, ZTD) the effect of the hydrostatic components (i.e. the Zenith Hydrostatic Delay, ZHD), the IWV can be estimated, provided that the surface

pressure $p_s$ and the water vapour weighted mean temperature of the atmosphere $T_m$ (Bevis et al., 1992) are known at the GPS site. In this paper, we use homogenously (re)processed (tropospheric) products released by the International GNSS Service (IGS, Dow et al., 2009) in November 2011. Details on the reprocessing analysis strategy are provided in Byun and Bar-Sever (2009). The reprocessing was applied to the complete IGS network, i.e. about 400 continuously operating GPS stations from 1995 until end 2007, and was identical to the operational processing strategy from then up to April 2011. As a result, we

dispose of a world-wide, nearly continuous and homogeneously reprocessed ZTD (and consequently IWV) dataset at 118 stations over a 16-year time period (see Fig. 1 and Table S1 of the supplementary material). The same IGS GPS ZTD dataset



has been converted into IWV and then analysed in Wang et al. (2016a) and Parracho et al. (2018). A second reprocessing campaign of the IGS network is foreseen, but its delivery date is still uncertain. Other GNSS networks, e.g. the EUREF (International Association of Geodesy Reference Frame Sub-Commission for Europe, http://www.euref.eu) Permanent Network (EPN) have recently performed a second reprocessing activity (Pacione et al., 2017) and are available up to more

recent dates (e.g. end of 2014 for the EPN repro 2), but are spatially more restricted (e.g. EPN repro 2 is limited to Europe), which does not fit the scope of this paper.

Although homogeneously reprocessed, the available IGS ZTD time series can contain discontinuities or break points due to instrumental changes (antenna, receiver and radome changes) or changes in the observation statistics (Vey et al., 2009 and Ning et al., 2016), which might have a (modest, see the discussion in Parracho et al., 2018) impact on the trends. The

homogenization of this dataset is out of scope for this paper, but is tackled in a working group of the past COST Action 1206 "GNSS4SWEC" (Advanced Global Navigation Satellite Systems tropospheric products for monitoring severe weather events and climate) and is continued under the umbrella of the International Association of Geodesy (IAG) Working Group (WG) 4.3.8 "GNSS Tropospheric Products for Climate". The adopted methodology of homogenization and some preliminary results of the participating homogenization algorithms are described in Van Malderen et al. (2017).

As already mentioned, for the conversion of the ZTD estimates to IWV values, auxiliary meteorological parameters at the GPS site, $p_s$ and $T_m$, are needed. $T_m$ can be either calculated from vertical profile data provided by radiosondes or reanalyses, or estimated from surface temperature ($T_s$) observations using a linear empirical relationship (e.g. Bevis et al., 1992). In this study, we use the surface pressure from the European Centre for Medium-Range Weather Forecasts (ECMWF) ERA-Interim reanalysis (Sect. 2.3) surface field. For the calculation of the weighted mean temperature, we make use of the ERA-Interim

pressure level data of temperature and specific humidity. For each of the retrieved meteorological parameters, the values from the four grid points surrounding the GPS station were horizontally interpolated, weighted with the inverse distance to the GPS station. To account for the height difference between the GPS station and the ERA-Interim surface grid points, we assume a standard temperature lapse rate of $-6.5$ K km$^{-1}$ (typical for wet adiabatic conditions) for the surface temperature altitude correction, the hydrostatic and ideal gas equations to adjust the surface pressure, and a constant dewpoint

temperature depression ($T$-$T_{dew}$) with height for the specific humidity adjustment (Deblonde et al., 2005). The vertical integration to retrieve $T_m$ starts with the ERA-Interim surface grid point values converted to the GPS station height, and continues higher up with the ERA-Interim pressure level data. We did not apply any time interpolation, so that, as the ERA-Interim reanalysis is only available at 0, 6, 12, and 18h UTC, the 5-minute resolution IGS repro 1 ZTD estimates are downsized to a 6h time resolution IWV dataset at these mentioned times. On average, the GPS IWV time series of our

sample of stations contain about 23% of gaps at this time resolution, with the lowest ratio (1.5%) reached for DRAO (starting on 1st Jan. 1995) and a ratio as high as 55% for SNI1 (also only covering the July 1996 – July 2010 time period).

Of course, the source choice of the auxiliary meteorological parameters needed for the ZTD to IWV conversion might have an impact on the resulting GPS IWV values and trends. Therefore, in the Appendix, we perform a sensitivity analysis by using different datasets (ERA-Interim, NCEP/NCAR reanalysis, SYNOP stations) and different calculation methods of $T_m$



(from the linear empirical relationship with the surface temperature or as outlined above). It turns out that, averaged out for all stations of our sample, the surface pressure has a larger impact on the IWV values than the surface temperature and weighted mean temperature, also on the derived IWV trends.

## 2.2 GOME/SCIAMACHY/GOME-2

Since the launch of the Global Ozone Monitoring Experiment (GOME) in 1995, satellite spectral measurements of moderate resolution in the red spectral range allow the IWV retrieval using differential optical absorption spectroscopy (Beirle et al., 2018, and references therein). The same retrieval strategy could be applied to the Scanning Imaging Absorption spectroMeter for Atmospheric CHartographY" (SCIAMACHY) spectra (August 2002 – March 2012), and to data gathered with an updated version of GOME, GOME-2 (since January 2007). Within the ESA GOME-Evolution project, a IWV "Climate" product was developed, merging, consistently, the time series of those three instruments to provide a monthly mean IWV $1° \times 1°$ global grid from July 1995 to December 2015. Consistency is reached by merging SCIAMACHY and GOME-2 observations to the larger GOME pixel size (320x40 $km^2$) and reducing the GOME-2 swath width to that of GOME and SCIAMACHY. Cloud effects, which are known to affect the retrievals (see e.g. Van Malderen et al., 2014 for the comparison with GPS IWV estimates), are also treated consistently for the different sensors by deriving the information required for air-mass factor correction and cloud masking directly from the spectral analysis.

The GOME-SCIAMACHY-GOME-2 time series are additionally homogenized by correcting for the offsets determined during sensor overlap periods. When validated with GNSS and ERA-Interim, mean biases of -1.0 and -0.65 mm, with RMS values of 4.3 and 3.4 mm are respectively found. A temporal stability of about 1% per decade is achieved for the GOMESCIA[1] climate product, as demonstrated in Beirle et al. (2018), and references therein, by comparison with the same independent IWV datasets. In this context, we want to mention that the satellites have a constant equator crossing time between 9h30 and 10h30 local time which has an impact on the comparison with the mentioned IWV datasets. All the details of this ESA GOME-Evolution "Climate" water vapour dataset can be found in Beirle et al. (2018). In this paper, we considered only the IWV monthly mean time series at the pixels closest to the GPS stations.

## 2.3 Reanalysis model output

We use primarily the ERA-Interim reanalysis here, not only as data source for the auxiliary meteorological parameters to convert the GPS ZTD to IWV, but also to provide directly IWV time series. The use of the NCEP/NCAR reanalysis is restricted to sensitivity analysis purposes because of its coarser spatial resolution.

---

[1] In the remaining of the paper, we use the acronym GOMESCIA to denote the three instruments GOME, SCIAMACHY, and GOME-2 at once.



### 2.3.1 ERA-Interim

ERA-Interim (Dee et al., 2011) is a global atmospheric reanalysis starting in 1979, continuously updated in real time, but with a 2006 release of the data assimilation system (not including ground-based GPS data). The system includes a 4-dimensional variational analysis (4D-Var) with a 12-hour analysis window, giving a temporal resolution of 6h. The spatial

resolution of the dataset is approximately 80 km ($0.75° \times 0.75°$) on 60 vertical levels from the surface up to 0.1 hPa. Today, this is the most state-of-the-art reanalysis product of the ECMWF, but an improved reanalysis product, ERA5, is foreseen to be completely available by the end of 2018.

We extracted the IWV values from the ERA-Interim surface field at the GPS station location, again by horizontal interpolation of the IWV values from the four surrounding grid points around the GPS station, weighted with the inverse

distance to the GPS station. Because the height of the GPS stations does not usually agree with the model topography used in ERA-Interim, a vertical interpolation of the ERA-Interim IWV to the GPS station height is required. We use the approach proposed by Hagemann et al. (2003), in which the adjustment is obtained by the (numerical) integration of the specific humidity $q$ over the height difference between the GPS station and the model surface in 30-m steps, which generally corresponds to a pressure difference smaller than 4 hPa. Here again, a constant dewpoint depression is assumed to height-

adjust the specific humidity.

The homogeneity of the extracted ERA-Interim IWV time series has been questioned recently by Schröder et al. (2016). By applying a statistical breakpoint detection test on the ERA-Interim anomaly differences relative to other reanalyses and to satellite IWV retrievals, these authors detected breakpoints in ERA-Interim that could be matched to changes in the observing systems or changes of the input to assimilation schemes. Also Ning et al. (2016) detected changepoints in GPS –

ERA-Interim IWV differences, which were validated using independent data sources (i.e., data obtained from nearby GPS and/or VLBI sites; the DORIS data were used for one site) and these changepoints were attributed to inconsistencies in ERA-Interim.

### 2.3.2 NCEP/NCAR

We additionally extracted IWV time series from the National Centers for Environmental Prediction/National Center for

Atmospheric Research NCEP/NCAR Reanalysis 1 (Kalnay et al., 1996), which has a coarser spatial resolution ($2.5° \times 2.5°$, or about 210 km) and only 17 vertical levels, but is also available 4-times daily. The data assimilation (3D-Var) and the global spectral model are identical to the global system implemented operationally at NCEP on January 1995, but with a higher horizontal resolution. The database has been enhanced with many observations not available in real time for operational use. We applied exactly the same methodology as for ERA-Interim to retrieve the IWV (and auxiliary

meteorological) values from NCEP/NCAR at the GPS station locations and heights. We only use this reanalysis dataset for illustration purposes here, as it has been shown that ERA-Interim has a better performance than NCEP for representing the IWV by evaluation with radiosondes, GPS and microwave satellite observations (e.g. Chen and Liu, 2016).



## 2.4 Teleconnection indices

Teleconnection indices are used in the stepwise multiple linear regression in Sect. 6.2 as explanatory variables for describing the long-term IWV time variability. Teleconnections are indicators for the low-frequency variability components (of e.g. pressure and circulation) in the atmosphere and ocean and describe the climate links between geographically separated regions. Climate teleconnections are present in observations that have been averaged in time over a period that is long enough to suppress the data-to-day weather fluctuations, but short enough to retain the seasonal-to-interannual component of climate variability. Teleconnection patterns can be extracted from correlation analysis and from the calculation of principal components, among other techniques (Nigam and Baxter, 2015).

Here, we used different sets of teleconnections indices, summed up in Table S2. We first mention 10 Northern Hemisphere atmospheric teleconnections: North Atlantic Oscillation (NAO), East Atlantic (EA), East Atlantic/Western Russia (EA/WR), Scandinavia (SCAND), Polar/Eurasia (POL), West Pacific (WP), East Pacific-North Pacific (EP-NP), Pacific/North American (PNA), Tropical/Northern Hemisphere (TNH) and Pacific Transition (PT). Their description, index calculation procedure and the timeseries are available at the Climate Prediction Center (CPC) website within the National Oceanic and Atmospheric Administration (NOAA): http://www.cpc.ncep.noaa.gov/data/teledoc/telecontents.shtml. These teleconnections reflect rather semi-permanent positions of the upper (e.g. 500 hPa) troughs and ridges due to the presence of quasi-stationary Rossby waves within the Northern Hemisphere extratropics. Almost all of them show a regional scale impact (e.g. EA, SCAND, NAO for Europe and WP, EP-NP, PNA – for the North American – Pacific sector). Further, Arctic (AO) and Antarctic (AO) oscillations represent annular structures of atmospheric circulation surrounding polar regions (and hence have also a close relation with the polar stratospheric vortexes), and consequently showing a hemispheric scale impact. Other used teleconnections represent rather oceanic or coupled ocean-atmosphere circulations, like Pacific Decadal Oscillation (PDO), Atlantic Multidecadal Oscillation (AMO), different ENSO indices: SOI, NINO 3.4, ONI etc. These indices reproduce the long-term variability (even on decadal scale) of sea surface temperature (SST) anomalies in certain regions of the oceans. The Northern Oscillation Index (NOI) reflects the variability in equatorial and extratropical teleconnections and is constructed using only sea level pressure data in defined regions. The quasi-biennial oscillation (QBO) is a quasi-periodic oscillation of the equatorial zonal wind between easterlies and westerlies in the tropical stratosphere with a mean period of 28 to 29 months.

## 3 Dataset comparison

Before analysing the time variability of the different IWV datasets, we first undertake a quick comparison between the monthly means of those datasets. We can only treat monthly means here, as the GOMESCIA climate product is only available with this time resolution. In Fig. 2, we show the (linear Pearson) correlation coefficients between the 3 different IWV datasets. From this figure, it is immediately obvious that GOMESCIA differs most from the other 2 datasets. Its





average R² with GPS is 0.865, while the average R² between ERA-Interim and GPS is equal to 0.975. This is not surprising, as GPS and ERA-Interim are "local" data, while the climate GOMESCIA IWV product is based on observations in GOME resolution, with an across-track width of 320 km, i.e. much coarser than the used 1°×1° grid. The worst correlations with GPS are in both cases obtained for island and coastal sites, where the spatial representation of the IWV field at the GPS site

location by the 1°×1° GOMESCIA ground pixel or the 4 surrounding ERA-Interim model grids can be questioned. It is therefore not a surprise that the correlation between GOMESCIA and ERA-Interim is higher, with an overall average of 0.918. Looking at the biases, we found that 70% of the GPS stations have a negative IWV bias with respect to ERA-Interim, while 60% of the stations have a positive bias compared to GOMESCIA. Especially for sites in Europe, Southeast Canada and East USA, GOMESCIA shows a dry bias with respect to GPS, which in turn is dry-biased compared to ERA-Interim. In

this context, it should be noted here that the GOMESCIA climate product was optimized for inter-instrumental consistency over time, not for accuracy (see Beirle et al., 2018). The standard deviations are smallest between GPS and ERA-interim, and largest between GPS and GOMESCIA. Here, the impact between the different observations times (= satellite overpass times) at the sites for GOMESCIA compared to GPS and ERA-interim should be highlighted. Furthermore, Schröder et al. (2018) pointed out that IWV retrieval techniques needing clear-sky conditions (like GOMESCIA) are marked by a larger

variability around the clear-sky class ensemble mean than all-sky retrievals (like GPS and ERA-Interim). The slopes of the linear regression lines between GPS and ERA-interim are very close to 1, but about 60% of the sites have slopes larger than 1. If we compare GOMESCIA with respect to GPS, the slopes are smaller than 1 for about 62% of the sites (but consistently not in eastern continental Europe). This has been ascribed in Van Malderen et al. (2014) to the cloud shielding effect when measuring with the GOMESCIA instruments during less cloud-free scenes. Finally, we also want to mention shortly that the

comparison of the GPS IWV monthly means with NCEPNCAR is very similar to the GPS-ERA-Interim comparison, although slightly worse. So, the use of the ERA-Interim meteorological variables for the ZTD to IWV conversion for the GPS retrievals is not the dominant factor for the good agreement between both IWV monthly mean datasets.

## 4 Frequency distribution

We first characterise the IWV field properties at the GPS stations by means of the frequency distribution of their time series.

This analysis enables us to make a classification of geographical regions with similar IWV properties, allowing us to carry out a regional analysis later in this paper. We exclude the GOMESCIA dataset here, as only monthly means are available, which might be problematic to compute significant frequency distributions. Foster et al. (2006) concluded that the histograms of precipitable water from radiosondes and zenith neutral delay (or ZTD) estimated by GPS are described at many locations by a lognormal distribution, contrary to e.g. surface temperature measurements that follow a Gaussian

distribution. In the appendix of their paper, they demonstrated that this observation is consistent with a theoretical derivation of the precipitable water lognormality, based on moisture flux. From an intuitive point of view, the lognormal distribution of IWV can be understood by the fact that IWV cannot be negative (zero lower bound) and might have a longer tail towards



higher values. The lognormal distribution function used in Foster et al. (2006) is described by three parameters: the median M, the geometric standard deviation (GSD) s, and a threshold parameter t allowing to have a non-zero lower bound (standard lognormal) or even a distribution bounded by an upper value t with the long tail tending towards zero (reverse lognormal). For their sample of 10 stations worldwide, Foster et al. (2006) could distinguish three categories: a traditional lognormal

distribution (for subtropical and temperate climate), a reversed lognormal form (in tropical oceanic environments) and a bimodal distribution, where seasonal (e.g. monsoon) or climatic (e.g. El Niňo) variations generate distinct precipitable water modes with rapid transitions between them.

In this work, we computed lognormal distribution functions (by a non-linear least squares fit) for the IWV distributions observed at each GPS sites with the same formula as in Foster et al. (2006). Examples of each category are given in Fig. 3.

We added an extra category, namely in between a lognormal and bimodal distribution, where there is one clear lognormal distribution which characterises the majority of the distribution, but where an additional, secondary lognormal distribution (most often at the higher IWV side, an upper mode) is needed to explain the overall frequency distribution satisfactorily (i.e. in terms of the chi-square goodness-of-fit statistic). We call it a "shouldered" lognormal distribution. The bulk of the shouldered or bimodal distributions are a combination of a standard lognormal distribution for the lower component and a

reverse lognormal distribution for the upper component, as in the examples in Fig. 3c and 3d. But the upper mode is more ambiguous, because often the overlap between the modes obscures the telltale asymmetry that would distinguish between them (also noted by Foster et al., 2006). The origin is clearly related to a large seasonal variation in the median values of the IWV, with the dry season characterised by the standard lognormal distribution with a low median, and the wet season characterised by a reverse lognormal distribution with high median value. As a matter of fact, if we consider deseasonalized

time series for those sites (obtained by subtracting the overall monthly means), the corresponding frequency distributions can be fitted by standard lognormal or even Gaussian distribution functions.

In Fig. 4a, we present the geographical distribution of those different classes of histograms for the GPS IWV time series. A very similar geographical repartition is achieved for the other datasets, unless mentioned otherwise. We immediately see that the large majority of the GPS sites have indeed lognormal distributed IWV time series, with a few exceptions, which have

Gaussian distributions (e.g. 2 sites in Antarctica which have very small IWV ranges). The NCEP/NCAR reanalysis has the highest number of Gaussian distributed IWV time series, which occur almost all at coastal or island sites (not shown here). It should also be noted that, for our sample, sites with a reverse lognormal distribution are very uncommon: only for the GPS IWV time series BOGT (Bogota, Colombia) and SAMO (Samoa Island, Polonesia), and also at Hawai, DGAR (Diego Garcia Island), KOUR (French Guyana) for the ERA-Interim time series. Another striking feature in Fig. 4a is the fact that

the occurrence of bimodal distributed IWV time series is restricted to the (sub)tropics (i.e. latitudes lower than about 30°). The dominance of a bimodal distribution for the Asian sites is linked to the seasonal behaviour due to the monsoon.

Australia, on the other hand, is characterised almost uniformly by a standard lognormal IWV distribution, see again Fig. 4a. For those stations, the distribution of each month can be described by a standard lognormal distribution, with slightly different values for the median and the GSD between months. Also the northern part of Europe (Scandinavia) has a standard





lognormal IWV distribution, while the rest of Europe (with the exception of some Mediterranean sites which are Gaussian distributed) have this shouldered lognormal distribution, which originates from the seasonal IWV variation as well.

North America is a mixture of standard lognormal and shouldered lognormal IWV distributions. The geographical distinction in this continent is further illustrated when considering the GSD values obtained when fitting one single lognormal distribution through the IWV frequency distributions (see Fig. 4b, but now for the ERA-Interim IWV dataset). Now it is obvious that the western part of North America is characterised by lower GSD values, while the central and east North American stations have higher GSD values (higher overall widths of their histograms) due to a long tail for higher IWV values in their distribution function. This tail can be ascribed to the large difference in the median values between the (dominant) lower and (weak, if present at all) upper mode of their distributions. This is illustrated by the frequency distributions of the deseasonalized IWV time series which mark the same geographical distinction: whereas the sites in western part of North America have standard lognormal distributions, the central and east North American stations are best fitted by shouldered lognormal distributions. For those latter stations, the IWV variability caused by weather or interannual variability seems to be more complex (multimodal) than if the seasonal variability is added.

The standard lognormal distributions for the Australian IWV time series (see PERT in Fig. 3a) are also described by low GSD values (see again Fig. 4b), and the overall width of the histograms is even lower than these of the North American west coast. In Europe, there seems to be a small increase in the GSDs of the histograms from west (maritime) to east (continental), although this might be mixed up with a latitudinal gradient in the GSD. In any case, the West European sites have frequency distributions with a broad peak (the two modes are close to one another, so that the overall distribution sometimes even seems to have a Gaussian shape, see e.g. GRAS in Fig 3c), with more distinct modes in the east of Europe (more pronounced tail). The Scandinavian sites have narrower IWV distributions around their peak, but are also strongly (positively) skewed, which explains the large GSDs.

To conclude, we confirm for a much larger sample the finding by Foster et al. (2006) that IWV follows a lognormal distribution. This is in particular the case if considering deseasonalized time series, although Gaussian distributions are then also more common. The seasonal variability - especially in the subtropics where seasonality is pronounced and distinct (monsoon), but also for most European sites - makes the frequency distributions more complex, favouring a multimodal (bimodal or shouldered lognormal) description. The opposite is however true for the central and east North American sites. Australian sites closely approximate a standard lognormal distribution, with or without considering seasonality. The analysis of the IWV frequency distributions therefore also aids in clustering sites with similar distributions in specific geographical regions or climate regimes.

## 5 Seasonal behaviour

To determine the seasonal cycle from the IWV, we calculate the long-term monthly means (=the mean for each month) restricting the monthly mean time series of the different datasets to subsets with identical starting and end dates for each site.



The geographical distribution of the amplitude and phase (= month with the maximum value) of the yearly cycle for the different datasets is shown in Fig. 5. It can be seen that overall, there is a good consistency among the three datasets, especially between GPS and ERA-Interim, with the seasonal cycle of GOMESCIA deviating most from the other two. This is also illustrated by the histograms of the retrieved amplitudes and phases of the different datasets (Fig. 6). The most

striking feature in those figures is that the IWV seasonal cycle for about 15 sites in the Northern Hemisphere peaks one month later in the GOMESCIA dataset with respect to the GPS and ERA-Interim datasets. The differences in amplitude are largest in the interval 8-14 mm, which is also the amplitude range for the bulk of the (Northern Hemisphere) sites in our sample. However, the correlation between the amplitudes is very good (0.98 between GPS and ERA-Interim, 0.81 between GPS and GOMESCIA).

Alternatively, we applied a linear regression model (see section 6.2 for more details) containing only harmonic functions (with periods of one year, 6 months, 4 months and 3 months) to determine the phase (=month with the maximum values) and amplitude (=magnitude of the harmonic functions with 12 month period) of the seasonal cycle. With this approach, the month delay in the seasonal cycle peak in the GOMESCIA dataset is even more pronounced, in particular with respect to the GPS dataset. Also the differences between GPS and ERA-Interim seasonal cycle properties are larger. This can be ascribed

to the rather broad distribution of the seasonal cycle properties of the GPS dataset, which is related to the poorer representation of the IWV time series by the set of harmonic functions: averaged over all sites, only 62% of the variability can be explained, leading to a correlation coefficient of about 0.764 between the dataset and the linear regression model of harmonics. These values are 81% and 0.895 for ERA-Interim respectively, and 71% and 0.831 for GOMESCIA. The worse parameterisation by harmonics for the GPS dataset can be partly explained, at least for some sites, by the presence of gaps in

the IWV monthly mean time series. However, in this context, it should be noted that a slightly higher number of sites were found to have a statistical significant contribution from a biannual cycle for ERA-Interim (79%) and GOMESCIA (75%) than for GPS (69%), which might also contribute to a better linear regression representation of course.

   To conclude, we found a very similar seasonal cycle for GPS and ERA-Interim. The GOMESCIA seasonal behaviour deviates more from the other two, especially for the Northern Hemisphere sites.

**6 Long-term time variability of the IWV retrievals**

Based on time series of global-mean monthly IWV anomalies derived from GPS, radiosonde and microwave radiometer data, Wang et al. (2016a) found that, for the period 1995-2011, IWV increases generally with time over both land and oceans during recent decades at a rate of around 0.26 mm decade$^{-1}$. Applying the statistical formalism developed in Weatherhead et al. (1998), and hence taking into account the effect of auto-correlation and variability as calculated from our IWV monthly

anomaly time series, we found that, on average, 38 years of monthly data are needed to detect a trend with this magnitude at a 95% confidence level with probability 0.90. This number is comparable with the number of years found by Roman et al. (2012) for GPS and a Global Climate Model (GCM), but for a 0.05 mm yr$^{-1}$ trend in IWV. For GPS and ERA-Interim time





series over Europe, Alshawaf et al. (2018) found that the number of years of daily data required to estimate a IWV trend above 0.3 mm/decade is between 28 and 40 years. As we have only 15 years available for most of the stations, our time series is too short to draw firm conclusions on the presence or magnitude of a trend. Instead, we concentrate on the difference between the trends calculated from the different datasets (Sect. 6.1) and on the interpretation of the time

variability by means of a stepwise multiple linear regression (Sect. 6.2).

## 6.1 Linear trends

We calculated linear trends as the slope of the linear regression line that was fitted (by minimising the least squares) through the monthly anomaly IWV time series. The standard error of the linear regression slope was used as an estimate of the uncertainty in the trend (Ross and Elliott, 1996). Additionally, to test the statistical significance of this trend, we applied

Spearman's test of trend (see also Ross and Elliott, 2001). The linear IWV trends for the different datasets are shown in Fig. 7 for the period January 1996 – December 2010. It can be seen that overall, the agreement in the sign and magnitude of the trends is rather good between the different datasets. Around 2/3th of the GPS stations have positive trends for the GPS time series (20 sites with statistically significant positive trends), whereas this is only 55% for GOMESCIA (and 9 sites with significant positive trends). For ERA-Interim, this percentage amounts to 60% (and 12 sites with significant positive trends).

The number of sites with statistically significant negative trends is for none of the datasets larger than 5. Comparing our mean IWV trends with the 0.26 mm decade$^{-1}$ GPS IWV trend quoted by Wang et al. (2016a), we found slightly lower rates of 0.19, 0.08 and 0.11 mm decade$^{-1}$ respectively for GPS, GOMESCIA and ERA-Interim. The correlation between the IWV trend values at the different stations is largest between GPS and ERA-Interim ($R^2$=0.66), but also large between ERA-Interim and GOMESCIA ($R^2$=0.63), and smallest between the two observational datasets ($R^2$=0.55). We should mention here

again that trend differences might be expected between all-sky IWV datasets (GPS and ERA-Interim) and clear-sky datasets (GOMESCIA), see e.g. Fig. 4 in Schröder et al. (2018).

The geographical consistency in the sign and magnitude of the trends between the different datasets is high for Europe (which agrees with the findings in Alshawaf et al. (2018) for a GPS dataset and ERA-Interim), where the datasets reveal an overall moistening (consistent with the strong surface warming for continental Europe shown in the lower right panel of Fig.

7). The drying above West Australia and the moistening over the Indian Ocean also seem to be consistent features among the three different IWV datasets. Another observation is that the three datasets do not show a clear geographical pattern of IWV trends for North America, especially GOMESCIA, but also the surface temperature trends of this continent are not fully geographically consistent. Qualitatively, the trend patterns shown here are in good agreement with Wang et al. (2016a), who, for the period 1995-2011, found positive trends (over land) along the coast of the Northeast USA and Eurasia as well as the

interior of Australia and Europe, and negative trends covering most of the eastern Pacific around the western coasts of the Americas. They also mentioned that the Atlantic and Indian Oceans are dominated by moistening trends. Parracho et al. (2018) pointed to the similarity of observed drying over Western Australia between their study (for the period 1995-2010)



and Wang et al. (2016a), but also reported on disagreeing trends over central Asia (moistening in Wang et al., 2016a) and a strong spatial variability of trends over the western part of USA in both studies.

## 6.2 Stepwise multiple linear regression of IWV time series

To understand the main drivers of the seasonal variability and the long-term time behaviour of the IWV, different
approaches are possible. For instance, climate models might be run and the underlying physical processes in the models might be studied and validated with IWV observations, an approach followed in Berckmans et al. (2018). In this paper, we try to fit the IWV time series of the different datasets by empirical "models", based on representations of circulation patterns (e.g. ENSO) and atmospheric oscillations (e.g. NAO), as described in Sect. 2.4.

### 6.2.1    Methodology

The following regression model is applied to the monthly means of the different IWV time series Y(t):

$$Y(t) = A_0 + A_1 t + \sum_{i=2}^{n} A_{seas,i} X_{seas,i}(t) + \sum_{j=0}^{m} B_j X_j(t) + \epsilon(t) \tag{1}$$

where $A_0$ is the intercept, $A_1$ is the annual trend, $A_{seas,i} X_{seas,i}(t)$ describe the seasonal cycle of the IWV, either by (i) using the long-term monthly mean for that specific month $X_{seas,2}(t)$ (n=2) or by (ii) using the sum of harmonic functions for $X_{seas,i}$ (i.e. $cos(2\pi t/12) + sin(2\pi t/12)$ and multiples of the cosine and sine arguments for 6 month, 4 month and 3 month periods, so that
n=8). For the choice of both options, see below. $X_j(t)$ are the explanatory variables (including the time series generated from teleconnection indices shifted with 1 to 6 months back in time, see below), and $B_j$ their respective coefficients, $m$ then denotes the number of candidate explanatory variables and depends on the geographical region (ranges between 103 and 194, see below). Finally, $\varepsilon(t)$ represents the residuals.

There are of course some obvious candidates for IWV explanatory variables like the surface temperature, surface pressure,
NAO, ENSO (see e.g. Wagner et al., 2006, Shi et al., 2018, Wang et al., 2018), Pacific Decadal Oscillation (PDO, see Shi et al., 2018), but we also made a primary assessment of the links between IWV and available teleconnection and various atmospheric and oceanic indices by (lag) correlation analyses between the index and the IWV. In practice, we calculated pairwise linear correlations between the indices on one side and both the (original) IWV time series and with the IWV lagging for 1, 2, 3...6 months on the other side, as e.g. circulation acts as a predictor and it always leads (or is simultaneous
with) IWV. An example of such a (lag) correlation analysis between IWV (and upper-tropospheric humidity) and the Niño 3.4 and PDO index is given in Shi et al. (2018). In our correlation analysis, we use the NCEP/NCAR reanalysis as the data source for the IWV time series at the stations, as not to favour one of the datasets (GPS, ERA-Interim, and GOMESCIA) that actually have been used for IWV in the stepwise multiple linear regression. The correlation analyses at the different sites revealed a regional pattern in how closely the IWV relates to an index. This is not surprising, as the distribution of the
frequency distributions (Sect. 4) and the linear trends (Sect. 6.1) clearly had a geographical imprint. So, we end up, for geographically distinct regions, with a list of candidate proxies which can be used in our multiple linear regression model.



This list is provided in Table S2 of the supplementary material. We also checked carefully if those explanatory variables are independent from each other, by looking at their linear correlation coefficients (these must be below 0.90). The total number of candidate explanatory variables (including their time series preceding the IWV time series with 1 to 6 months) in the multiple linear regression ranges between 103 (Australia) and 194 (North America).

Then, in the first step of our multiple linear regression, for each site, we calculated single linear regressions of each of the explanatory variables (selected for the geographical region the site belongs to) against the IWV time series (also lagging for 1 to 6 months) and sort them according to their explained variance. Then, we subsequently incorporated into the regression function (1) the variables, starting with the ones with the highest explained variance. The statistical significance of each included variable was tested by means of a t-test of the regression coefficient. Variables with a significance level lower than 5% are discarded.

### 6.2.2 General results

For two sites (DUBO at Lac du Bonnet, Canada, and HOB2 at Hobart, Australia), we show the multiple linear regression fits to the IWV time series, retrieved respectively by GPS and GOMESCIA in Fig. 8, together with the residuals (observations minus the fit) in the lower panels. Those two sites are typical examples of a good fit (DUBO, for which the fit explains 98.64% of the variability or with a correlation coefficient of 0.993 between the observations and the fit) and a bad fit (HOB2, with an explained variability of 54.55% and a correlation coefficient of 0.739). The graph also shows that, despite the very high percentage of explained variance of the fit for the DUBO GPS IWV time series, a significant positive trend is still present in the residual time series (although the annual trend was not retained as a significant explanatory variable in the multiple linear regression). To fit the seasonal cycle of the IWV time series shown in Fig. 8, we made use of the long-term monthly means in our Eq. (1), instead of using harmonic functions. In general, we found that the explained variances (or correlations coefficients) by the fits are very similar, and this for the different datasets, between both proxies used to describe the seasonal behaviour. But, these explained variances are reached with, on average, fewer other explanatory variables when using the long-term monthly means. Apparently, when using the harmonic functions, a part of the seasonal behaviour present in the time series still has to be explained by other variables, especially by the surface temperature and precipitation time series (see further). In Sect. 5, we already noted that larger differences in distribution of the phase and amplitude of the seasonal cycle between the datasets occur when the seasonal cycle is parameterised by harmonics instead of using the long-term monthly means. For those reasons, we focus here on the results of the multiple linear regression analysis with the long-term means as proxy for the seasonal cycle.

The explained variability by the linear regression model fits for the two observational datasets is shown in Fig. 9, and for ERA-Interim in Fig. 10. Let us first concentrate on the differences between the different datasets used. Overall, the best fits are obtained for the ERA-Interim output (on average 90%), and the "worst" fits for the GOMESCIA dataset (on average 85.5%), with the GPS time series there in between (88.9%). On average, the higher ERA-Interim explained variance is also obtained by including a higher number of explanatory variables (8.2) in the linear regression than for GPS (8.0) and





GOMESCIA (6.8). Given a total number of candidate explanatory variables between approximately 100 and 200 (see above), there does not seem to be an overkill in the statistically significantly retained variables. As, globally, the most important explanatory variables (next to the long-term means) are the surface temperature, precipitation, surface pressure and tropopause pressure, (both in their frequency of occurrences and in the explained variability), the highest explained

variability of the ERA-Interim dataset might be therefore related to the fact that the surface temperatures and pressures at the site locations are calculated from the ERA-Interim reanalysis. Moreover, ERA-Interim data has been used to convert the GPS ZTD to IWV, which might also explain the good linear fits of the GPS IWV time series with dominant proxies calculated from ERA-interim. However, we also applied the stepwise multiple linear regression to the GPS IWV time series converted from ZTD using NCEP/NCAR, and nearly identical explained variability was obtained for this GPS dataset. The

lower value for the GOMESCIA dataset seems to be linked with the significant lower percentage of stations for which the precipitation and the surface pressure are included in the linear regression (resp. around 40% and 25%), in comparison with the other two datasets (resp. above 70% and around 40%). The surface temperature on the other hand is included as an explanatory variable in the GOMESCIA IWV linear regression for about two third of the stations, which is the highest frequency of the datasets (resp. 57 and 62% for GPS and ERA-Interim), but the average explained variability is lower (71%

versus 80 % for GPS and ERA-Interim).

The dominance of the surface temperature and precipitation as "explanatory variables" for the IWV time series is of course no surprise, since both the spatial (pattern) and temporal correlation between time series of those variables have already been highlighted in different studies (e.g. resp. Trenberth et al., 2005 and Allan and Soden, 2008). The already mentioned Clausius-Clapeyron equation clearly links those variables (see also Pall et al., 2007, and Lenderink and van Meijgaard, 2010,

Chen and Liu, 2016), although regional moisture divergence/convergence can cause significant departures from Clausius-Clapeyron rates. The surface temperature and precipitation become even more dominant explanatory variables (in frequency) when harmonic functions are used as proxies for the seasonal cycle instead of the long-term monthly means. As the seasonal cycle is best described by the latter, this strengthens our hypothesis that the surface temperature and precipitation account for some part of the remaining seasonal variability in the IWV time series. To be complete, we mention that also the surface

pressure is used for a larger number of stations when using the long-term means in the multiple linear regression equation, but the contrary is true for the tropopause pressure.

Having a closer look at Figs. 9 and 10, it is clear that the explained variability by the multiple linear regression fits is marked by a geographical dependency, which is identical for the different datasets used. The best fits are obtained for Canada (sites with latitudes higher than 45°N), East USA, East Asia and Europe. This might probably be explained by the larger

availability of northern hemisphere teleconnection patterns (both for the Atlantic and Pacific Oceans) as explanatory variables, in comparison with southern hemisphere counterparts. The quoted regions are also the regions with the highest number of retained explanatory variables in the linear regression, next to the West USA. The reason for the lower explained variability for these Californian stations is not clear at the moment. It should also be noted that relatively high explained



variability is obtained for the Antarctic stations for the ERA-Interim dataset only, while Parracho et al. (2018) pointed out some spurious IWV trends in ERA-Interim over this continent compared to GPS and MERRA reanalysis model output.

### 6.2.3 Geographical footprint of explanatory variables

In this section, we focus on the interpretation of the IWV time variability in terms of the dominant explanatory variables for specific geographical regions. "Dominant" means here that the explanatory variables (or the lagged IWV time series by one to six months) were retained by the multiple linear regression, by means of the t-test, in more than half of the stations in that specific region, and this for the three different datasets considered here (GPS, GOMESCIA, ERA-Interim). A schematic summary of the forthcoming discussion is given in Fig. 10. We consider specific geographical regions because the candidate explanatory variables used in the regression analysis were tied to different continents as a result of our (lag) correlation analysis. However, for North America for instance, a further subdivision was needed to group the stations for which the multiple linear regression kept similar explanatory variables. This subdivision is hinted at by the geographical patterns found in the analysis of the frequency distributions (Sect. 4).

The **North American** sites have a clear west-east separation in terms of geometric standard deviations of their single lognormal distribution (Fig. 4b), probably related to the special orography due to the Cordilleras in the west. Overall, the regression fits of western sites explain a smaller amount of the IWV variability (Figs. 9 and 10) than for the central and eastern sites. Especially the *Californian* sites are less fitted by the multiple linear regression. However, there is large consistency in the used explanatory variables among the three datasets for California: for almost all stations, the North Oscillation Index (NOI), the North Pacific (NP) and West Pacific (WP) indices were retained. Associated temperature and precipitation patterns of these indices (see e.g. http://www.cpc.ncep.noaa.gov/data/teledoc/ telecontents.shtml) point to the possible impact they might have on the IWV variability in California. Also the vertical component of the Eliassen–Palm Flux (EPF) through the tropopause at 100 hPa and averaged over 45–75 degrees north, as a proxy for the Brewer–Dobson circulation (e.g., Brunner et al., 2006), is consistently included in the regression for about half of the stations for the 3 datasets. For about half of the sites, a part of the IWV variability is also explained by the precipitation (for GPS and ERA-Interim) or the surface temperature (for GOMESCIA). The lifting motion of the large scale westerlies from the Pacific over the windward slopes of the Cordilleras lead to moisture convergence and IWV variation strongly depending on cyclonic/frontal activity coming from the North Pacific.

The most significant explanatory variable for the *Canadian* sites (we also include Fairbanks in Alaska here, and the Northeast USA sites NLIB, WES2 and GODE) is the tropopause pressure, for the three IWV datasets. It therefore seems that the IWV variability is linked with the time variability of the vertical extent of the troposphere, in which the water vapour resides. In those areas, plains prevail (except the Appalachian mountains, where no stations of our sample our located). The surface temperature, precipitation and Tropical/Northern Hemisphere (TNH) index are frequently used (and explain a large fraction of the IWV variability) for GPS and ERA-Interim, but are less important or insignificant for GOMESCIA. Other important explanatory variables for the entire Canada are the NP index and the Arctic Oscillation (AO) index, which arise in





resp. about two third and half of the sites for the IWV datasets. The NAO index is present in about 40 to 50% of the stations for the three datasets. The already mentioned differences between the western (Vancouver area) and central/eastern sites are somewhat reflected in additional dominant explanatory variables: the surface temperature and the East Atlantic/West Russia (EA/WR, whose impact on the IWV variability is questionable here) for the Vancouver area, but the surface pressure for

central/east Canada & USA.

The few sites in **Latin America** are widely distributed over the continent, but dominated by coastal regions. As such, there is no consistency between the frequency distributions, the correlations between datasets, the trends and explained variability by the multiple linear regression fits. The only explanatory variable that is for almost all sites used in the three datasets is the precipitation. For Latin America too, the retained indices deviate mainly between GOMESCIA and the other two datasets.

For GPS and ERA-Interim, the ENSO (Southern Oscillation index) and Pacific Transition (PT) index are frequently (above 60%) used as explanatory variables, a role that is fulfilled by the EPF for the GOMESCIA dataset. It should however be noted that the PT circulation is only active for the months August and September. For the other proxies, we did not find any geographical consistency within this continent for their use (Pacific west coast vs. Atlantic east coast, north vs south, etc.).

The **European** sites have comparable geometric standard deviations of their histograms compared to the west coast USA

sites, but the histograms are mainly shouldered lognormal distributions, except for the high latitude sites, which have standard lognormal distributions (Fig. 4). This might be due to the similar conditions as the American case: the storm track from the North Atlantic transports heat, moisture and momentum (stronger in winter), giving high interannual and interseasonal variability, but also high intra-seasonal variability. There also seems to be a latitudinal gradient in the retained explanatory variables in the linear regression: the tropopause pressure is present at nearly all higher latitude ($>55°$) sites for

the three datasets, but almost completely absent for the Mediterranean sites. Also the East Atlantic/West Russia (EA/WR) index is much more prominent for the higher latitude sites. On the other hand, the Tropical/Northern Hemisphere (TNH) index, which only occurs for the winter months, contributes only for the lower latitude European sites, as is confirmed by correlation maps between TNH and precipitation (see http://www.cpc.ncep.noaa.gov/data/teledoc/tnh_pmap.shtml). However, for every European site is the surface temperature the most important explanatory variable, even more frequently

present than the long-term means. The precipitation is also clearly linked with the IWV variability for a large majority of the sites (in particular for GPS and ERA-Interim). Other important proxies are the East Atlantic (EA) and Polar/Eurasia (POL) indices, for which (especially the EA) the associated temperature and precipitation patterns are situated in Europe (see again http://www.cpc.ncep.noaa.gov/data/teledoc/telecontents.shtml). The EPF and the Atlantic Multidecadal Oscillation (AMO) arise for around 60% of the stations with large explained variability for GPS and ERA-Interim, but only respectively for 40

to 48% of the sites for GOMESCIA. The seasonal variation is a strong component in the time series of these explanatory variables, which might contribute to their selection here. Quite surprisingly, the NAO index is present in only one third of the sites as explanatory variable, although its relationship with precipitation is well established in Europe (e.g. Casanueva et al., (2014) and references therein).



5 out of 6 sites in **East Asia** are characterised by a bimodal frequency distribution (Fig. 4), but nevertheless are very well fitted by a multiple linear regression equation (Figs. 9 and 10). The retained explanatory variables might hence give a hint for the processes responsible for the bimodal frequency distribution. In this context, we mention that when using harmonic functions instead of the long-term means to account for the seasonal behaviour in the multiple linear equation, this bimodality is, for the three datasets, represented by the cosine and sine functions with period 6 months for 2 sites only, and with small values of the explained variability. However, whatever the proxy we used for the seasonal variations (harmonics or means), we found that the precipitation and surface pressure time series at the sites are primarily used in the linear regression to account for the variability in the IWV time series (including the bimodality of the frequency distribution), especially for GPS and ERA-Interim. But also the ENSO and the NP index are, not very surprisingly, important proxies for the IWV time variability for all the three datasets. The EPF plays a role in exactly 3 out of the 6 stations for the three datasets, but with larges values for the explained variability. For the two observational datasets, the Pacific Decadal Oscillation (PDO) index occurs for half of the sites as an explanatory variable, which is expected since it is the leading principal component of monthly SST anomalies in the North Pacific Ocean.

The **Australian** sites have a standard lognormal frequency distribution and two trends patterns were consistently observed among the three datasets: a drying in western and middle Australia and a moistening elsewhere (Fig. 7). The explained variability by the linear regression varies a lot over the continent (Figs. 9 and 10), although the surface temperature and pressure and precipitation were all included for the bulk of the stations. Australia is the continent for which the long-term monthly means are less frequently (around 60%) used to fit the seasonal cycle. Furthermore, the EPF and the PT index are important contributors for the three datasets, the ENSO (Southern Oscillation Index, SOI) is only for the GPS dataset not so frequently used (only in one third of the stations), although it is expected to be important for the water vapour field in this region (see e.g. Wang et al., 2018). The selection of explanatory variables does not give a clear discrimination between the drying west and middle Australia and moistening east, so we cannot give more insight to the analysis in Parracho et al. (2018), where the role of the dynamics was assessed by considering the wind field at 925 hPa.

Finally, we consider the sites at and around **Antarctica**, where mostly a moistening occurs (except for ERA-Interim for some sites). We mention here that no precipitation time series are available for the Antarctic sites, so that it could not be included in the multiple linear regression equation, but we do not expect an effect of IWV on precipitation. For all sites, among the three datasets used, the surface temperature is used as explanatory variable in the regression analysis, and explains the bulk of the remaining variability, after the long-term mean (which is only not used for 2 sites for ERA-Interim). For about half of the sites, the Atlantic Multidecadal Oscillation (AMO) is also an important proxy in terms of the explained variability. To a lesser extent, the same is true for the EPF, but only for the ERA-Interim and GOMESCIA dataset. The link with AMO might be the signal coming from the North Atlantic SST through the Tropical Atlantic SST Dipole (Trades) and affecting the Southern Atlantic subtropical anticyclone intensity, which is one of the factors influencing the De La Plata cyclogenesis (in Argentina, Uruguay), from which cyclones come to the shores of East Antarctica. Alternatively, because both the AMO and EPF have a strong seasonal signal in their time series, their selection might be triggered by remaining



seasonality in the time series, which could not properly be accounted for by the surface temperature and the long-term means.

## 6.3 Discussion

Although the length of the time series is limited, statistically significant linear trends (according to Spearman's test) have been found for a number of sites (ranging from 13 to 25 for GOMESCIA and GPS respectively). Therefore, a linear trend was added as a candidate explanatory variable in the stepwise multiple linear regression for all geographical regions. However, only in few sites (MATE, CFAG and FALE for ERA-Interim and ANKR for GPS), the linear trend was retained as explanatory variable by the statistical test. Because the number of sites for which a statistical significant trend is still present in the residuals (as e.g. for the site DUBO in Fig. 8) is at most 11 (for GPS), the stepwise multiple linear regression fit is mostly able to capture the long-term trend without using the linear trend proxy. Moreover, whereas the majority of the sites have positive trends in their IWV time series, especially for the GPS and ERA-Interim datasets (see Sect. 6.1), the residual time series after applying the multiple linear regression show an equal amount of positive and negative trends (GPS and ERA-Interim) or even a higher amount of sites with a negative trend (GOMESCIA).

To identify which variables also account for the long-term variability (or linear trend) of the IWV, we first focus on the dominant explanatory variables: the surface temperature, the precipitation, and the surface and tropopause pressure, all determined locally at the site. As was already shown in Fig. 7, a surface warming occurred during the 1996-2010 time period for most of the sites (55%), and this warming was statistically significant for 18 sites, which is the double of the amount of sites for which a significant cooling was calculated (mostly situated in the western part of Northern America). The precipitation on the other hand decreased in the majority of the sites (60%), but only in 9 sites significantly (mostly again in West Canada & USA), against 5 sites with a significant increase in the precipitation. The surface pressure time variability is similar to this of the precipitation (decrease in about 70% of the stations, of which 9 have a significant decrease in surface pressure), and the tropopause pressure time behaviour resembles this of the surface temperature (increase in about 60% of the stations, but only significant in 3 sites, compared to 7 sites with a significant decrease in tropopause pressure). As a matter of fact, more or less independent of the dataset used, we found that, on average, for about 70% of the cases for which the mentioned (local) explanatory variables are present in the multiple linear regression of the sites, the linear trend sign of the explanatory variable's term (coefficient multiplied with its time series) is in agreement with the linear trend sign of the IWV time series of the same site. For the precipitation and the surface pressure, this percentage of trend sign agreement is even between 70 and 75% for the three datasets used, while lower percentages are found for the surface temperature for the GPS time series (around 65%) and for the tropopause pressure (around 60%) for both the GPS and GOMESCIA time series. So, from this analysis, it seems that those 4 local explanatory variables, when retained in the regression, are more or less equally responsible for explaining the trend sign of the IWV time series.

Also other, global, explanatory variables used have a trend in their time series for the considered 1996-2010 time interval. In particular, a significant decrease in the NAO index and significant increase in the NOI (also when preceding the IWV for 6




months) are determined; all the other explanatory variables are marked by insignificant trends in their 1996-2010 time series. The NAO coefficients in the linear regression are mostly positive for Northern America (the NAO decrease leads to an IWV decrease) and mostly negative for Europe (the NAO decrease increases the IWV), while the NOI coefficients are all negative for the Californian sites (weaker Pacific subtropical anticyclone, higher probability for the storm track to pass the Californian

coast), but positive when this index precedes the IWV time series for 6 months. So, for some sites, the trend present in the linear regression term of an explanatory variable is in line with the IWV trend, but for other sites has the opposite sign. It should be clear that, as the resulting IWV trend in the linear regression term is the sum of all the contributions of the retained explanatory variables, the precise identification of the main contributor to the IWV trend is almost impossible.

## 7. Conclusions

For a sample of 118 sites, globally distributed over the world, we analysed the IWV datasets, retrieved homogeneously by GPS and GOMESCIA in the period 1996-2010, making use of the ERA-Interim model output, both for the conversion of the GPS ZTD data to IWV and as a supplementary IWV source. Due to the 6h time resolution of ERA-Interim, both the GPS and ERA-Interim IWV datasets are available 4 times a day, but the GPS dataset contain on average about 23% of gaps. The GOMESCIA "Climate" product is provided as monthly means.

Next to a quick comparison of the IWV monthly means retrieved by the different datasets, we focused in this paper on different aspects: (i) the frequency distributions of the 6-hourly GPS and ERA-Interim IWV datasets, (ii) the amplitude and phase of the IWV seasonal cycle(s) present in the different datasets, (iii) the linear IWV trends in the 1996-2010 time period, and (iv) a stepwise multiple linear regression of the different IWV dataset making use of explanatory variables like surface temperature and pressure, precipitation, but also teleconnection patterns and climate and oceanic indices like ENSO, NAO,

North Pacific index, etc. The main findings are summarised here.
The GOMESCIA IWV dataset differs most from the other two datasets, and the worst agreement, between the three datasets, is obtained for island and coastal sites, where the spatial representation of the IWV field at the site location by the GOMESCIA ground pixel (320 km east-west) and surrounding ERA-interim model grids can be questioned. In addition, the climate product was optimized for consistency over time, not for accuracy (see Beirle et al., 2018).

As in Foster et al. (2006), we found that the frequency distributions of the IWV time series are best fitted with lognormal distributions, although for (sub)tropical sites and sites in East Asia two distinct lognormal distributions are needed, probably related to the monsoon and ENSO, giving rise to a bimodal lognormal density distribution. Sites in Europe and about half of the sites in Northern America have histograms that are best represented by a leading lognormal distribution for the lower component, added with another, reverse lognormal distribution for the upper component. Australia, on the other hand, is

characterised almost uniformly by a standard lognormal IWV distribution. The distribution of the geometric standard deviations of the lognormal distributions is also geographically very consistent.




According to the Weatherhead et al. (1998) formalism, the length of the time series is, in combination with the auto-correlation and variability of the IWV monthly anomalies, too short to detect the quoted trend of 0.26 mm decade$^{-1}$ by Wang et al. (2016a). If we however concentrate only on the sign of the trends among the different datasets (not on the statistical significance), then we can conclude that a positive IWV trend occurs over Europe and the Indian Ocean, while a negative trend takes place at West Australia. The trend sign pattern above North America is less consistent, especially due to GOMESCIA.

A stepwise multiple linear regression fitted to the IWV monthly means revealed that, after accounting for the seasonality by harmonics or long-term means, the bulk of the variability is explained by the surface temperature and precipitation. The precipitation is retained in the largest number of sites as explanatory variability, but with a lower explained fraction of the variability than the surface temperature. Of course, the link between surface temperature and moisture (IWV and precipitation) is well established due to the Clausius-Clapeyron equation. The other site specific explanatory variables, the surface pressure and tropopause pressure, are also important contributors to the multiple linear regression, the latter seems to be more common at high-latitude sites. To identify which of those explanatory variables accounts most for the trend sign of the IWV time series is not straightforward: as a matter of fact, we found that each of those variables results in a regression term with equal trend sign as the IWV trend sign for about 70% of the cases, and this for the three datasets.

But also the footprint of more global/regional teleconnection patterns or climate/oceanic indices was detected in the IWV time series of a geographical subset of sites. The North Pacific index is a dominant explanatory variable for West USA, Canada and East Asia, while the West Pacific index occurs for a majority of the West USA sites. The Pacific Transition index arises for Australia, and to a lesser extent for Latin America. The link of the ENSO with the IWV variability is well established over East Asia, but also present in Australian and Latin American sites. For the West USA, the North Oscillation index is also very dominant, and the Arctic Oscillation in Canada. Somewhat surprisingly, the NAO index could be linked to the IWV variability at Canada as well, but not in Europe. In this last continent, the East Atlantic, Polar/Eurasia, Tropical/Northern Hemisphere and Atlantic Multidecadal Oscillation indices are important contributors. The presence of the Eliassen-Palm flux, a proxy for the Brewer-Dobson circulation, in a large number of sites, worldwide, with large explained variability, is less understood, but might be related to the fact that this proxy has a strong seasonal signal in its time series, which might explain a portion of the remaining seasonality in the IWV time series, not properly taken into account by e.g. the long-term IWV means, surface temperature, and other variables.

The fact that, starting from a large selection of candidate explanatory variables and allowing for their time series to precede the IWV time series by 1 to 6 months (i.e. 100 to 200 variables), results in a considerable small amount of retained explanatory variables (around 7 to 8 on average) with a reasonable geographical footprint, and mostly consistent among the three datasets used, highlights the applicability of this method, even on the level of the time series of individual sites. Despite the remaining inhomogeneities in some of the datasets (e.g. due to instrument changes at some GPS sites, due to changes in the data assimilation sources in ERA-Interim and when combining the measurements of the individual instruments to build




up the GOMESCIA time series), this study also points out that combining three completely different IWV datasets enables to characterise the IWV variability on a regional scale.

## Appendix: : Impact of the auxiliary meteorological data for GPS IWV retrievals

The formulas used for the GPS ZTD to IWV conversion (Van Malderen et al., 2014) contain 2 meteorological variables: the surface pressure and the weighted mean temperature. Of these two, the impact of the surface pressure on the IWV is largest: a 1 hPa change in $P_s$ gives an IWV change of 0.36 mm, whereas a 1 K change in $T_m$ leads to an IWV change in the range 0.05 to 0.20 mm, depending on the ZTD and $P_s$ values. If the Bevis et al. (1992) linear $T_m$-$T_s$ regression is used ($T_m = 70.2 + 0.72\ T_s$), a 1°C change in $T_s$ brings an IWV change of 0.03 to 0.15 mm, as the $T_m$ changes with 0.72 K for a 1°C $T_s$ change. It should however be noted that the surface temperature also appears in our formula to convert the surface pressure from the height of the meteorological station to the GPS station's height.

Different data sources, either observational or reanalysis datasets, exist for the surface pressure and temperature and weighted mean temperature. To assess the sensitivity of the retrieved IWV values and trends on those different data sources, we analysed the comparison between two IWV datasets with different sources for these parameters. The different surface data sources used here are the network of meteorological stations – only available for about 40 IGS stations of our sample – and the ERA-Interim and NCEP/NCAR reanalysis datasets. The weighted mean temperature is either obtained from the surface temperature by the Bevis et al. regression, or calculated from the vertical profile data supplied by the reanalysis datasets. In  Table A1, we present the means of some statistical properties – weighted by the number of observations for each station – describing the (trend) differences between IGS IWV datasets that disagree only by one or more of these auxiliary meteorological parameters. Table A1 confirms the larger impact of the surface pressure than the surface temperate and weighted mean temperature on the IWV biases, but also on the derived IWV trends. Moreover, the slope of the linear regression (with correlation coefficient 0.84) between the $P_s$ and IWV biases between the different corresponding datasets for the 40 IGS stations is equal to the -0.34, confirming the acceptable data quality of the pressure observations at the retained stations. Although large differences exist between the surface and mean temperatures taken from the synoptic observations or ERA-Interim, see [c] to [e] in Table A1, the resulting IWV biases remain very small, as well as the IWV trend differences. The values for [d] can be compared to the study undertaken by Wang et al. (2016b). For our sample of data, the slopes of the linear regression between the $T_s$ and $T_m$ biases with the IWV biases of these datasets are 0.03 and 0.05 respectively (with correlation coefficients equal to 0.90 and 0.66 respectively), which are the lower end values of the ranges mentioned in the previous paragraph.

The effect of the used reanalysis dataset (see [f] in Table A1) on the resulting IWV biases and trends are comparable with the numbers for the effects of the temperatures ([c] to [e] in Table A1), except for the absolute bias. Apparently, the larger mean differences between the surface pressures of the two reanalyses (not shown in Table A1), compared to the pressure differences between ERA-Interim and SYNOP ([b] in Table A1), are partially compensated by the differences in mean





temperatures between the two reanalyses. The same reasoning might be applied to explain the smaller differences in IWV biases and trends between datasets generated completely with SYNOP observations and ERA-Interim on one hand ([g] in Table A1), compared to changing only the source data of the surface pressure ([b] in Table A1). Finally, we note that the largest IWV biases are obtained when comparing the ERA-Interim IWVs with the IWVs retrieved from the IGS ZTDs (see

5 [a] in Table A1). This is not surprising, as these two are the most independent IWV datasets shown in the table. It also illustrates the added value (negative or positive, this has been investigated in the paper) of the IGS data to the reanalysis data. The largest IWV trend differences are obtained for the comparisons [a], [b], and [g] in Table A1, implying that the IWV trends are driven by the trends in the surface pressures and the trends present in the ZTD time series themselves. The impact of the trends in the temperatures seems from the point of view of this sensitivity analysis (relying on the weighted

10 means of the entire set of stations) less determinative for the IWV trends. In any case, a correlation between the IWV trend differences and trend differences in $P_s$, $T_s$, and $T_m$ cannot be found, as was the case for the IWV biases.



**Table A1: Weighted mean values of the statistical parameters describing differences computed between two different IWV datasets, demonstrating the impact of different variables or dataset choices. The abbreviations used to describe the different IWV datasets are explained in Table A2. An asterisk denotes that the comparison was done for only 40 stations, i.e. those stations with a meteorological station within 30 km distance. The numbers in italic mark the weighted mean values of the parameters between the two different databases of the meteorological variable whose impact on the IWV is studied.**

| | bias (mm) | abs bias (mm) | SD | R² | trend (mm/dec) | abs trend (mm/dec) |
|---|---|---|---|---|---|---|
| [a] | **Tm ERA**: influence IGS data | | | | | |
| **IWV ERA** | -0.313±1.011 | 0.669±0.818 | 3.807 | 0.962 | -0.102±0.457 | 0.322±0.338 |
| [b] | **Tm ERA**: influence $P_s$ | | | | | |
| **TmERA P synop***  | -0.200±0.614 | 0.301±0.570 | 7.575 | 0.961 | -0.290±0.752 | 0.353±0.724 |
| | *0.031±0.611* | *0.357±0.494* | *0.786* | *0.989* | *0.052±0.401* | *0.277±0.292* |
| [c] | **T ERA P synop**: influence $T_s$ | | | | | |
| **TP synop*** | 0.009±0.025 | 0.020±0.017 | 0.029 | 1.000 | 0.013±0.244 - | 0.107±0.219 |
| | *0.267±0.622* | *0.503±0.447* | *2.878* | *0.979* | *0.197±0.879* | *0.502±0.744* |
| [d] | **Tm ERA**: influence Bevis et al. regression | | | | | |
| **TP ERA** | 0.044±0.092 | 0.069±0.075 | 0.025 | 1.000 | -0.004±0.031 - | 0.018±0.025 |
| | *0.508±1.583* | *1.235±1.108* | *8.099* | *0.804* | *0.101±0.213* | *0.188±0.140* |
| [e] | **Tm ERA P synop**: influence $T_s$ and Bevis et al. regression | | | | | |
| **TP synop*** | 0.026±0.071 | 0.051±0.055 | 0.052 | 1.000 | 0.001±0.248 | 0.110±0.221 |
| [f] | **Tm ERA**: influence reanalysis dataset | | | | | |
| **Tm NCEP** | -0.034±0.286 | 0.168±0.233 | 0.154 | 0.995 | -0.015±0.144 | 0.083±0.118 |
| [g] | **Tm ERA**: observational vs. reanalysis dataset | | | | | |
| **TP synop*** | -0.073±0.403 | 0.223±0.342 | 4.705 | 0.971 | -0.210±0.667 | 0.259±0.649 |

**Table A2: Nomenclatuur for Table A1.**

| name | $T_m$ | $P_s$ |
|---|---|---|
| **Tm ERA** | calculated from ERA-Interim | ERA-Interim |
| **Tm ERA P synop** | calculated from ERA-Interim | synoptic stations |
| **T ERA P synop** | converted from $T_s$ ERA-Interim with Bevis regression | synoptic stations |
| **TP ERA** | converted from $T_s$ ERA-Interim with Bevis regression | ERA-Interim |
| **IWV ERA** | IWV directly taken from ERA-Interim surface fields | |
| **TP synop** | converted from $T_s$ from synoptic stations with Bevis regression | synoptic stations |
| **Tm NCEP** | calculated from NCEP/NCAR | NCEP/NCAR |




**Author contributions**

RVM did most of the analysis, drafted the manuscript and designed the figures. EP, under the supervision of CB, provided the IGS dataset and frequently interacted and discussed with RVM during the analysis. GS did the correlation analysis of the teleconnection patterns with IWV, helped with describing the teleconnection patterns and with the interpretation of the
results of the stepwise multiple linear regression analysis. SB and TW provided their GOME/SCIAMACHY Climate IWV time series and gave very useful suggestions for the analyses described in especially sections 4 and 5. HB interacted with RVM and EP during the analysis and gave feedback on the draft.

**Acknowledgements**

The research has been undertaken in the framework of the Solar-Terrestrial Centre of Excellence (STCE), funded by the
Belgian Federal Science Policy Office. R.V.M., E.P. and H.B. are members of the Solar-Terrestrial Centre of Excellence. NCEP Reanalysis data and LIM SST Anomalies Forecast data and provided by the NOAA/OAR/ESRL Physical Science Division and CIRES CU, Boulder, Colorado, from their website at https://www.esrl.noaa.gov/psd/. The multiple stepwise linear regression code used here is based on my_stepwise, IDL program code, written by Axel Thomas, Institute of Geography, Mainz University, Germany. This work has been presented at various workshops organised within the
framework of the European COST Action ES1206 GNSS4SWEC (GNSS for Severe Weather and Climate monitoring; http://www.cost.eu/COST_Actions/essem/ES1206), so we are indebted to many colleagues contributing with their feedback and discussions!

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





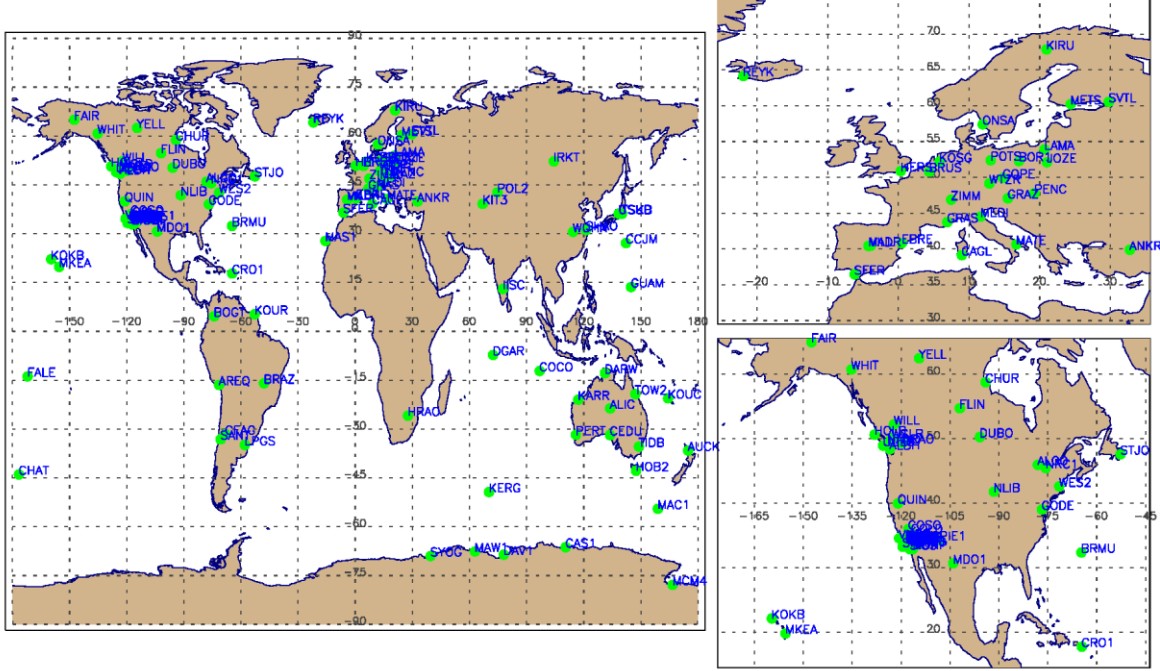

Figure 1: Maps of the selected 118 IGS stations with data in the period 1995/1996 – March 2011 (see text).

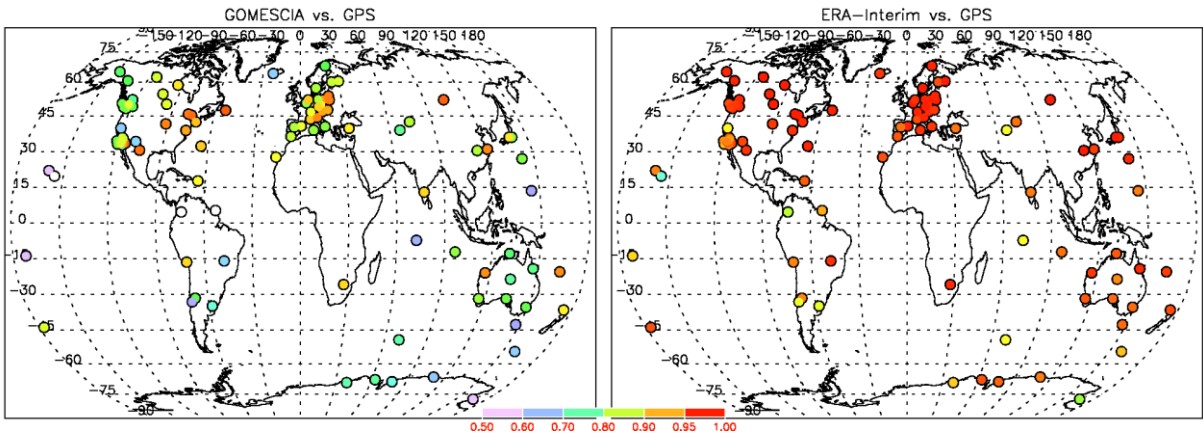

5    Figure 2: Correlation coefficients between the monthly means of GOMESCIA and GPS (left) and ERA-Interim and GPS (right).





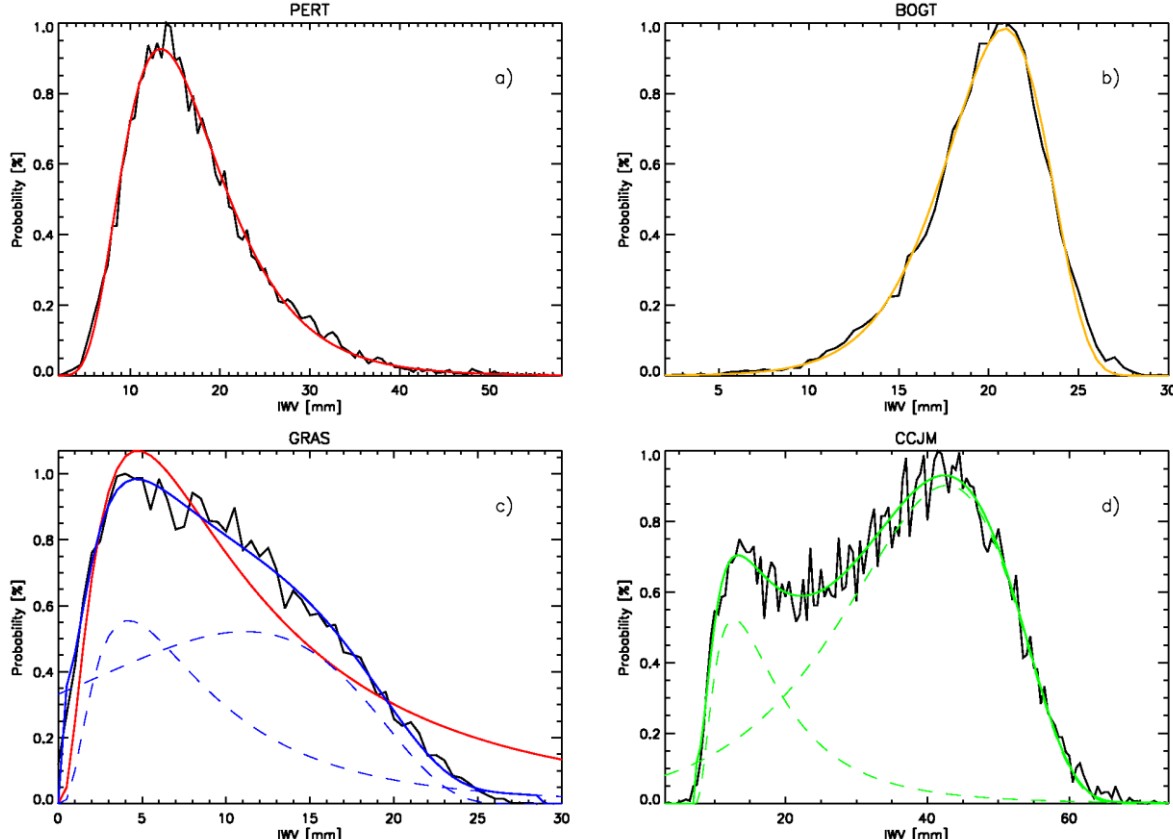

**Figure 3: Examples of the different categories of frequency distribution functions for the GPS IWV distribution at 4 GPS sites: a) the standard lognormal distribution at PERT (Perth, Australia), b) the reverse lognormal distribution at BOGT (Bogota, Colombia), c) the shouldered lognormal distribution (in blue, with the two contributing lognormal distributions in dashed blue) at GRAS (Caussols, France), and, for illustration, the best fit of a single lognormal distribution in red, and d) the bimodal lognormal distribution at CCJM (Ogasawara, Japan) with its contribution lognormal distributions in dashed lines.**





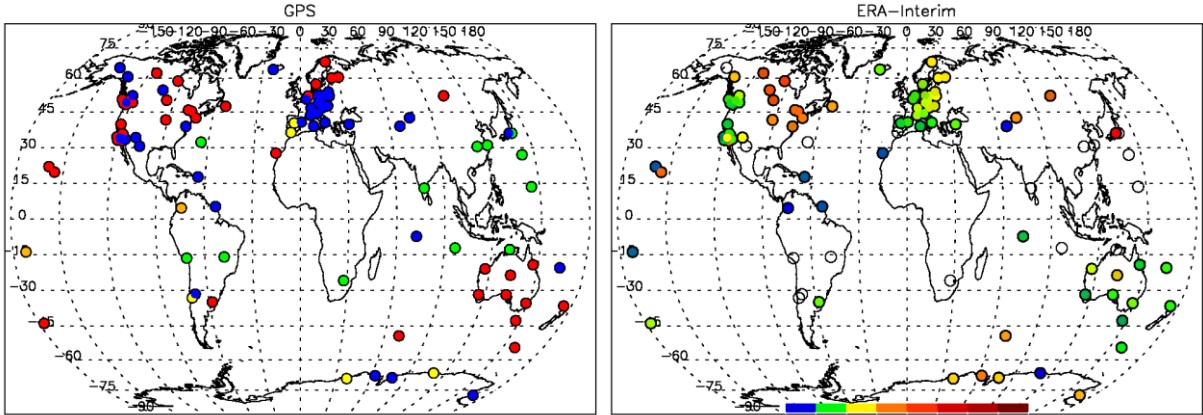

**Figure 4: a) Classification of the GPS IWV time series according to their frequency distributions: Gaussian (yellow), standard lognormal (red), reverse lognormal (orange), shouldered lognormal (blue), and bimodal (green). Those colours correspond to the colours used in Fig 3 for the different categories. b) Distribution of the geometric standard deviation (GSD) of a single lognormal distribution fitted through the ERA-Interim IWV histograms. The sites with unfilled circles have bimodal distributions.**




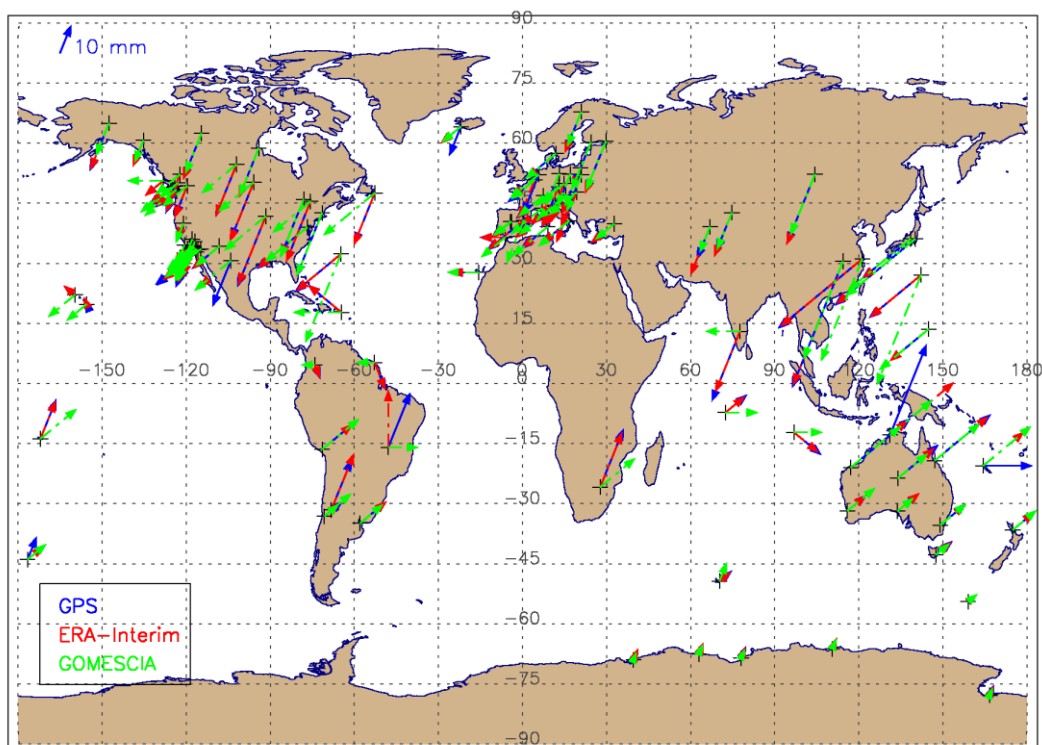

**Figure 5: Geographical distribution of the amplitude (length of the arrows) and phase (direction of the arrows like a clock: 1h=Jan, 2h=Feb, 3h=Mar, etc.) of the seasonal cycle of the monthly mean IWV time series of GPS (blue), ERA-interim (red) and GOMESCIA (green). A seasonal cycle of 10 mm amplitude in IWV is illustrated, as reference, by the length of the arrow in the upper left corner.**

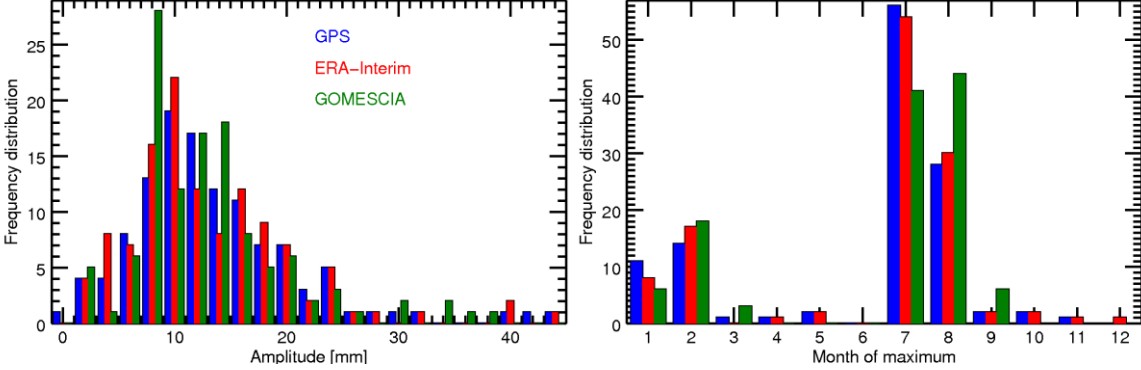

**Figure 6: Frequency distribution of the amplitudes (left) and phases (= month of the maximum IWV value, right) of the seasonal cycle in the monthly mean IWV time series of GPS (blue), ERA-Interim (red) and GOMESCIA (green).**



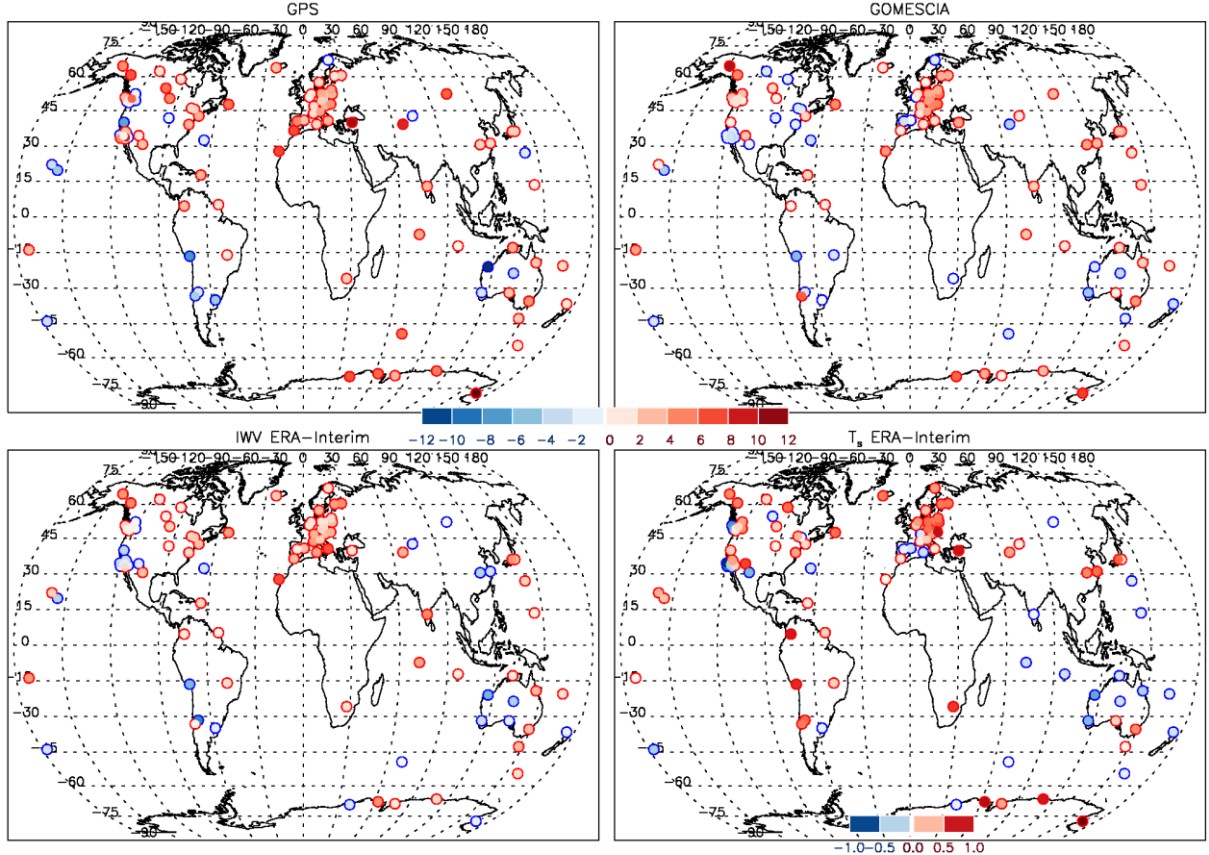

**Figure 7: IWV trends [%/dec] for GPS (upper left), GOMESCIA (upper right) and ERA-Interim (lower left) for the period Jan 1996 – Dec 2010. For illustration, the lower right panel shows the ERA-Interim surface temperature trends for the same period in °C/dec.**

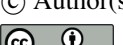



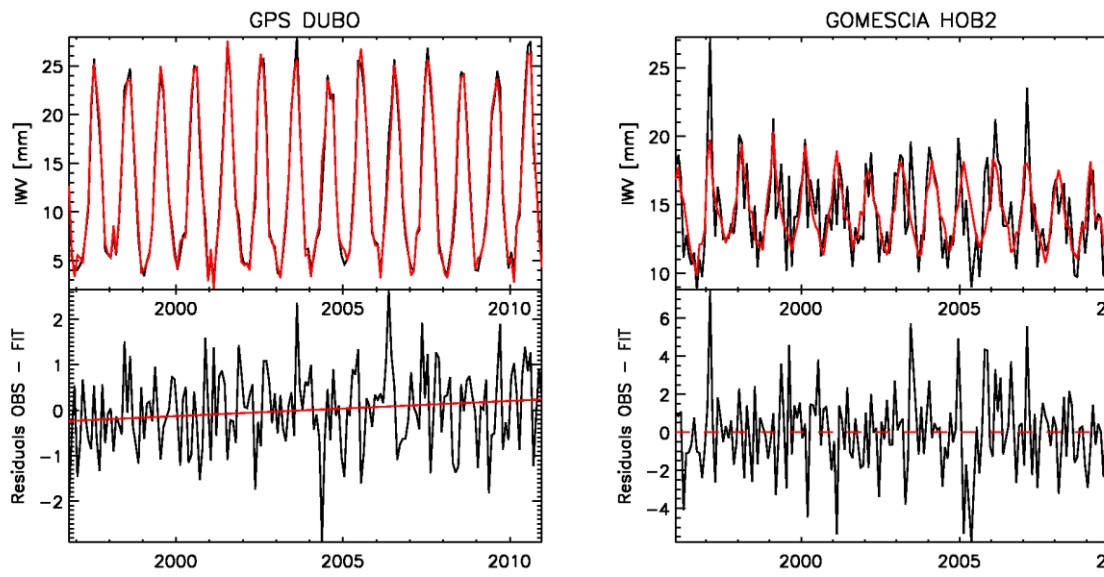

**Figure 8: Examples of the stepwise multiple linear regression fits (in red) to (left) the GPS monthly mean IWV time series of DUBO (Lac Du Bonnet, Canada) and (right) the GOMESCIA IWV time series of HOB2 (Hobart, Australia). The lower panels show the residuals between the observations and fitted time series, with the linear fit to the residuals in red (positive trend in both cases, but only significant (full line) for DUBO). The used proxies here were the long-term means, P$_{trop}$, AMO (especially preceding with 5 or 6 months), Prep, PT (preceding 1 month), P$_{surf}$, NP, EAWR (preceding 1 month), AO (preceding 5 months), NOI (preceding 6 months), WP, NAO, SOI (preceding 5 months) for DUBO (explaining 98.64% of the variability) and the long-term means, P$_{trop}$, WHWP (preceding 4 months), Prep, AOD (preceding 4 months) and P$_{surf}$ for HOB2 (explaining 54.55% of the variability). See Table S2 for explanations of the proxies.**

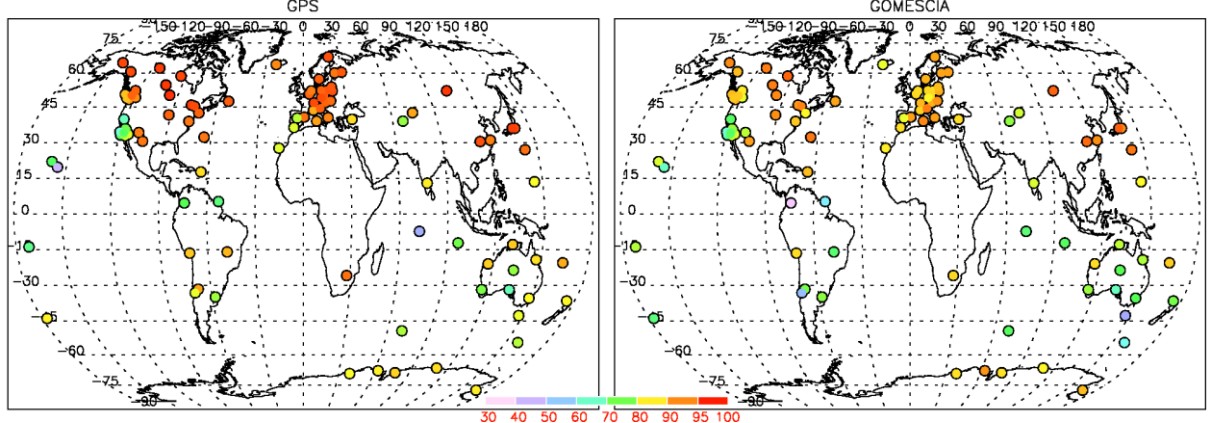

**Figure 9: Explained variability by the final multiple linear regression fit of the GPS (left) and GOMESCIA (right) IWV time series.**





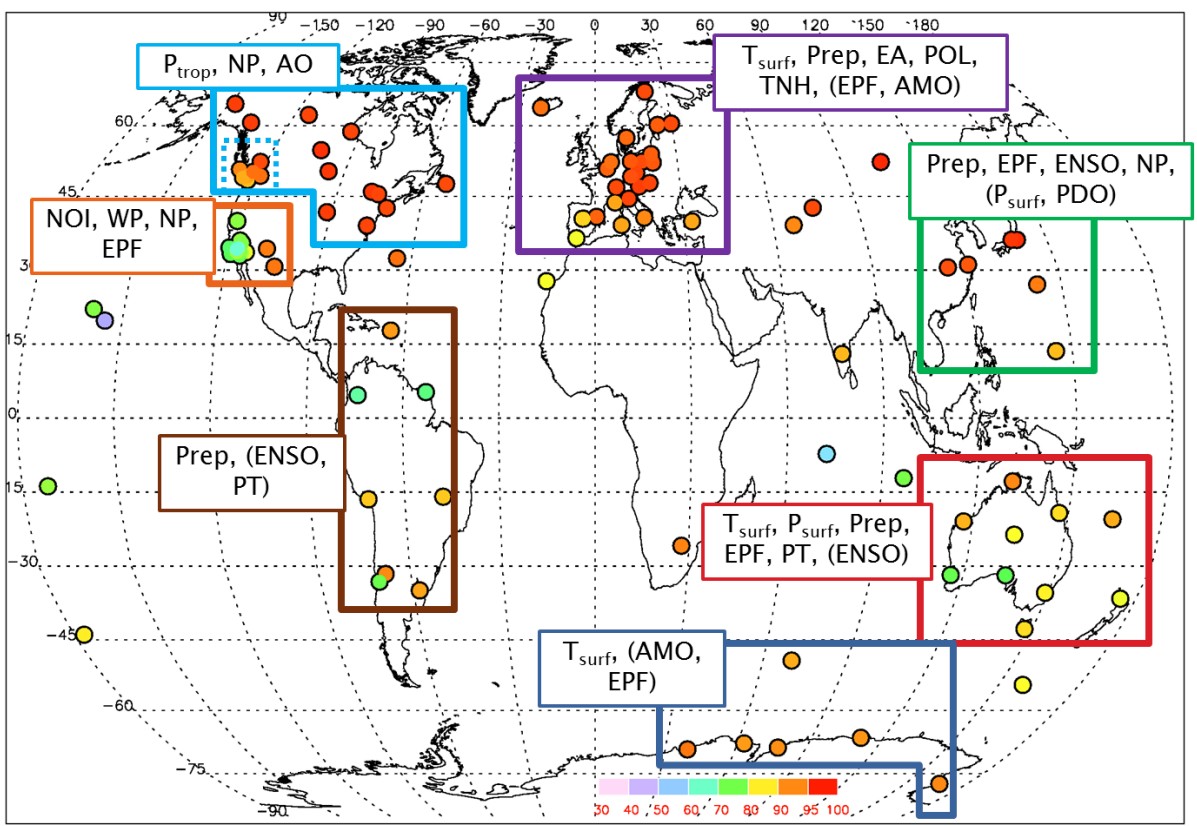

**Figure 10: Summary of the statistically significant explanatory variables that account for the variability of the IWV time series of the three datasets for at least half of the stations for different geographical regions. Between brackets: explanatory variables not entirely meeting these criteria, but still dominant in two datasets and/or in a considerable amount of the stations. The colours of the dots indicate the explained variability by the multiple linear regression of the ERA-Interim IWV time series.**