# Peer review of "Interpreting the time variability of world-wide GPS and GOME/SCIAMACHY integrated water vapour retrievals, using reanalyses as auxiliary tools"

_Atmospheric Chemistry and Physics, 2018_

## Referee Comment (RC1) · Anonymous Referee #1 · 25 Jan 2019

Review of the manuscript

"Interpreting the time variability of world-wide GPS and GOME/SCIAMACHY integrated water vapour retrievals, using reanalyses as auxiliary tools" by Roeland Van Malderen, Eric Pottiaux, Gintautas Stankunavicius, Steffen Beirle, Thomas Wagner, Hugues Brenot and Carine Bruyninx

The manuscript present the results of a study focussing on the variability of integrated water vapour across the globe as provided by two different datasets (sub-daily GPS

and monthly-mean GOMESCIA estimates).

I find the topic important and the results really quite interesting, and I liked the summary Figure 10. My main concern is about the way the scientific questions are addresses (too vague), the motivation behind the approaches proposed (not discussed), a lack of precision and explanations in numerous parts of the manuscript as well as too many speculations. As a result, this is sometimes confusing, so that I recommend at least a major review.

Some examples of lack of precision: 1) the teleconnection indexes are presented with no logical link, for instance, you do not explain your choice, nor systematically provide the associated fluctuation time scales; as a result this is confusing; 2) The section describing the GPS dataset is quite long but I could not find information on how the authors dealt with missing data; 3) in the regression analysis, I am not sure surface pressure and atmospheric temperature are taken at the site location or over the region to which the site belongs; 4) sentences such as "for (sub)tropical sites and sites in East Asia two distinct lognormal distributions are needed, probably related to the monsoon and ENSO" why ENSO? 5) in Figure 4, there are more tones of colours than indicated in the figure caption (e.g. pale green versus darker green, same for yellow to green). 6) "Comparing our mean IWV trends with the 0.26 mm decade-1 GPS IWV trend quoted by Wang et al. (2016a), we found slightly lower rates of 0.19, 0.08 and 0.11 mm decade-1", 0.10 is NOT slightly lower than 0.26!

It may be a good idea to separate the manuscript in at least two parts, so that it may be easier to present proper presentations and discussions of the results (which are very interesting and numerous). Including more material in appendix may help too.

Also, I think it would be excellent to take advantage of the global dataset (GOMESCIA) to add maps covering the whole globe rather than only presenting maps with results shown at GPS sites (data are sparse in large areas of the world and very clustered in others).

On another subject, while I can understand your choice for a regression method where you test very numerous potentially explaining variables, I think that you should discuss more how your results compare or not with other studies and mechanism (I understand that it is difficult because there is a lot in the manuscript). Also, I did not understand your motivation for including the linear trend as an explanatory variable while it was already taken into account in Eqn (1) and finally did not explain much more.

Below I present some specific comments on the first part of the manuscript. This is not exhaustive, but I hope they can help the review process and help you to revise the following sections and conclusions accordingly for a second round of review.

Specific comments

Abstract: you need to be more precise about the time periods and time scales of analysis.

Page 1, line 14, "IWV variability": please precise at which time scale and over which period, and the IWV sampling time step.

Page 1, line 18, "on average": this is too vague, on average over what?

Page 1, line 20-21, "the seasonal behaviour and the long-term variability are fitted together": this is not exactly what I understood. Rather, you aim at reconstructing the time series of monthly-mean IWV from the mean annual cycle, linear trend and explanatory variables.

Page 1, line 25, "long term trend": please precise, i.e. linear trend over the period [year1,year2]

Page 1, line 26: variableS

Introduction

In the last paragraph of the introduction, you present the work presented but I could not really find a clear presentation of the question(s) you want to address. In my opinion,

an analysis of a new type is not a good enough motivation per se.

Page 2, line 4, 'on local scale': are you referring to mesoscale here? I would rather say that "At all scales" rather than local scale because precipitation for instance is not simply related to "local water vapour" alone; it typically involves larger, e.g. synoptic atmospheric circulations as well. In addition to the diabatic processes you mention, I would add radiative processes. I would remove "Of course".

Page 2, line 7, 1st sentence: a reference is needed there.

Page 2, line 26, "have the potential to be used for climate change analysis, which is the subject of this paper": I do not agree about this statement, the results of this study are more focused on interannual variability (and trend over the 20-year-long period). The present study provides very valuable information about IWV variability in space and time. In time, it goes from the annual cycle to short inter-annual variability to trends over periods of 20 years or less. In my opinion, 20 years is too short to provide robust information on climate change. It is less than the time interval typically used to compute climate mean (30 years, e.g. see http://www.metlink.org/climate/depth-climate-met-office/).

Page 2, line 31: add 'generally' before 'occur' as this is to my knowledge not strictly true for all geographic locations.

Page 3, line 2: remove 'can' as you precisely provides number who illustrate it.

Page 3, line 4: remove "of course" and replace "can be" by "are" as existing studies allow you to be more affirmative.

Page 3, line 6: I am not sure of what you mean by "autocorrelation" here.

Page 3, line 6, about ENSO: it seems to me that, more precisely, the relatively large magnitude of the signal induced by ENSO events at inter-annual scale affects trends computed on periods of 10-20 years. It would be good to reformulated a bit the sentence to be more informative.

Page 3, lines 16-17: I think it is "on one hand", not "at one hand". More generally would be useful to check English throughout the manuscript. Avoid expressions such as "not surprisingly" or "of course" when you do not provide explanation nor references.

Page 4, lines 7-14, about homogenization: as the dataset in use here is not homogenized, I think this paragraph is unnecessary long. You could mention in the conclusion "for extension/improvement of this study, the use of a future homogenized dataset (as described in Van Malderen et al. (2017)".

Page 4, presentation of GPS data processing: you may consider moving part of it in supplementary material.

Page 6, about ERA5: this is not used, so the whole sentence is useless. It could be used in the future only, so potentially, you could mention it in the perspectives.

Page 6, lines 16-22: I am wondering why you mention this with so much details without linking this to your study.

Page 6, line 32: it seems to me that prior to Chen and Liu (2016), other studies such as by Bock et al. already extensively evaluated ECMWF and NCEP products.

Page 7, presentation of teleconnection indices: these indices are presented without much logic, you must re-write this section in a way that motivates your choice, and explain more over which time scales/regions they are relevant (and add somewhere the precise coordinates of the regions presented in the last figure).

Page 8, section 3: 1) how did you deal with missing GPS data? 2) It is not well suited to use the word 'bias' as you do not have a reference dataset here.

Page 8, "We exclude the GOMESCIA dataset here, as only monthly means are available, which might be problematic to compute significant frequency distributions": I do not understand what you mean. It seems you are mixing statistical robustness and time scale issues. Pages 8, 9, 10 and Figure 4: there are more tones of colours than indicated in the figure caption (e.g. pale green versus darker green, same for yellow

to green). Please clarify. Also, I would like to see the full maps obtained with ERA-I, NCEP as well as monthly-mean GOMESCIA IWV values. This would allow assessing the representativeness of the results obtained at the sites, and provide a clearer picture than emerging now at the end of this section. It is also very difficult to see the results over Europe because circles are overlapping each other.

Page 11, Figures 5 and 6: A discussion of the geographical patterns shown in Figure 5 is missing. In Figure 6, I am wondering about how this graph was made: did you consider all the stations? I guess the uneven location of the stations is playing a large role in the shape of these histograms. I would like to see the same graph with all the global GOMESCIA dataset.

Page 11, section 6: Given the content of this section, I suggest that you modify the title (especially remove "long-term"). You could rather emphasize the idea of the trend over [year 1, year 2] and inter-annual variability in this time window. You could probably shorten your discussion of the statistics, and it would be clearer if you could add a few words about decadal and multi-decadal variability.

Page 11, lines 2-3: "As we have only 15 years available for most of the stations, our time series is too short to draw firm conclusions on the presence or magnitude of a trend." Then, just below you compute trends. The way it is written is very confusing. I suspect you mean an "expected climatic trend" in the first instance.

Figure 8: It is important to add the residual obtained when removing the mean annual cycle and the linear trend to the time series, in order to be able compare it with its magnitude with the magnitude of the residual that you show.

I have more specific comments to come on a revised version of the manuscript where you would have taken into account my general and specific comments, as I think this manuscript should be published (perhaps as a two-part paper).

2018.

---

## Referee Comment (RC2) · Anonymous Referee #2 · 5 Feb 2019

**General Comments:**

In this paper an analysis of integrated water vapour (IWV) data from three different sources (ground based GPS data, GOME/SCIAMACHY/GOME-2 satellite data and ERA Interim model data) is presented. Beside intercomparisons of the three data sets major topics of the paper are the analysis of time series and the association of variabilities in IWV with 'explanatory variables', e.g. physical / dynamical processes described by teleconnection indices.

[Figure]

This is an interesting approach which is well presented, but some aspects need further clarifications. Especially, the method of stepwise linear regression needs to be explained in more detail. Which terms of eq. (1) are fitted in the individual steps (and in which sequence), which corrections are applied to the data sets used in each step? Especially, it should be clarified if/how e.g. seasonal variability and trends of the 'explanatory variables' is considered. Have these data been de-seasonalised before the fit? If not, how are correlations with the linear trend and seasonal terms in eq. (1) handled? It seems that e.g. linear trends are sometimes not fitted at all because they are covered by explanatory variables variations - how is this decided? Does the sequence of fitted variables matter?

Maybe it would help to add e.g. for one example a plot of the different fit steps with the different time series used in each step (not necessarily for all explanatory variables, but for the major ones); this could possibly be an extension of Fig. 8.

Another aspect is the selection of relevant explanatory variables for the fit. I would assume that many of the variables (e.g. temperature, pressure, precipitation) are strongly correlated - it needs to be explained in more detail how this is considered in the fitting procedure as well as in the interpretation of the results.

**Specific Comments:**

1. p. 6, l. 28–29:
   'The database has been enhanced with many observations not available in real time for operational use.'

   What is meant with this? Which database/observations do you refer to? Did you use/produce a dedicated version of NCEP/NCAR data? Please clarify.

2. p. 9, 1st paragraph:
   As I understand, the difference between lognormal and reversed lognormal distribution is that there is either a (non-zero) lower or upper limit for the IWV. Is there

a physical reason why a lognormal distribution occurs for subtropical and temperate climate and a reversed lognormal form in tropical oceanic environments?

3. p. 9, l. 10ff. and Fig. 3:
Since the 'shouldered' lognormal distribution is a new category I suggest to add also an example plot for this in Fig. 3.

4. p. 11, l. 5–6:
'the IWV seasonal cycle for about 15 sites in the Northern Hemisphere peaks one month later in the GOMESCIA dataset with respect to the GPS and ERA-Interim datasets'

Is there an explanation for this?

5. p. 12, l. 7–8:
'We calculated linear trends as the slope of the linear regression line that was fitted (by minimising the least squares) through the monthly anomaly IWV time series.'

Please explain how these anomalies are calculated (e.g. by subtraction of harmonics or long-term monthly means).

6. p. 12, l. 16–18:
Is here an explanation for the differences in GPS trends from this study and Wang et al. (2016a)? If the same data set is used (as stated in section 2.1), why are there differences?

7. p. 14, 1st and 2nd paragraph:
Are seasonal variations considered in the checks for independency of explanatory variables? Where does the limit of 0.90 for the linear correlation coefficients come from (in the section on linear trends a correlation of $R^2$=0.66 is considered as large).

In this context, and also for the regression tests used to determine the significance of the different explanatory variables: Have the data sets been deseasonalised before the comparisons? Is the preparation of variables for this test consistent with their later use in the fit (see also general comment)? Is the fact that 'Variables with a significance level lower than 5% are discarded' the reason that in the fits instead of 100-200 variables only 6-8 (see end of this page) are considered?

Please explain.

8. p. 14, l. 17–19:
   'a significant positive trend is still present in the residual time series (although the annual trend was not retained as a significant explanatory variable in the multiple linear regression).'

   This is unclear. If (as in eq. (1)) the linear trend is fitted, why is there a remaining trend in the residual?

9. p. 14, l. 23–25:
   'part of the seasonal behaviour present in the time series still has to be explained by other variables, especially by the surface temperature and precipitation time series'

   As mentioned in the general comments, it has to be clarified how seasonal variations are considered in each of the fitted time series and how this is in line with the seasonal terms in eq. (1).

10. p. 14, l. 23–25:
    'the NAO index is present in only one third of the sites as explanatory variable, although its relationship with precipitation is well established in Europe'

    If NAO index and precipitation are closely related, why can these be considered as independent explanatory variables?

11. p. 18, l. 26:

    'we do not expect an effect of IWV on precipitation'

    Why not? Please explain.

12. p. 18, l. 10–13:

    'Moreover, whereas the majority of the sites have positive trends in their IWV time series, especially for the GPS and ERA-Interim datasets (see Sect. 6.1), the residual time series after applying the multiple linear regression show an equal amount of positive and negative trends (GPS and ERA-Interim) or even a higher amount of sites with a negative trend (GOMESCIA).'

    Obviously, there are different ways to determine trends. As I understand, trends from section 6.1 originate from a simple linear regression (and probably a to be defined seasonal correction, see above). It seems that the multiple linear regression trends mentioned here are those remaining after subtracting the effects of explanatory variables without a linear trend fit (although a linear term is given in eq. (1)). This should be clarified.

13. p. 19, l. 25–27:

    'the linear trend sign of the explanatory variable's term (coefficient multiplied with its time series) is in agreement with the linear trend sign of the IWV time series of the same site'

    It is not clear what is meant with 'trend sign of the explanatory variable's term' and how this is derived. Do you fit a linear trend to the explanatory variable's term? Please explain.

14. p. 20, l. 25–27:

    'the precise identification of the main contributor to the IWV trend is almost impossible'
This sounds rather pessimistic. In most cases there will not be one single con-
tributor to the IWV trend. The analysis presented in this paper at least shows for
certain regions the main contributors, and this is a very useful result which could
possibly even be the basis for further investigations (see suggestion below).

15. p. 22, l. 14:
    The 'meteorological station' is only introduced in the next paragraph, should be
    explained before.

16. p. 22, l. 14:
    'the slope of the linear regression (with correlation coefficient 0.84) between the
    Ps and IWV biases between the different corresponding datasets for the 40 IGS
    stations is equal to the -0.34, confirming the acceptable data quality of the pres-
    sure observations at the retained stations.'

    It is not fully clear to me what has been done here and how the derived numbers
    are to be interpreted. Do you refer here to the begin of the appendix 'a 1 hPa
    change in Ps gives an IWV change of 0.36 mm'? Table A1/A2 list three different
    Ps sources - to which do you refer here?

17. Table A1:

    (a) Please define "abs bias" and "abs trend". Probably these are the bias and
        trend of the absolute differences? Why are these absolute numbers relevant
        in relation to the interpretation of the multilinear fit?

    (b) What exactly is meant with case a)? Is this a comparison between the IGS
        and the ERA Interim IVW data?

    (c) What is meant with 'the two different databases of the meteorological vari-
        able whose impact on the IWV is studied'? For example, I interpret case b)
        as a comparison between IWV results based on Ps from ERA Interim and

Ps from synoptic stations (this should be the non-italic numbers). What are the italic numbers in this case?

18. Figure 4:
Why do you only show Classification for GPS (a) and GSD for ERA Interim data (b)? I assume these plots should be available for both data sets.

19. Figure 8:
It should be clarified that no linear trends are fitted, only proxies (if this is the case here).

**Technical Corrections:**

1. p. 2, l. 16:
differntial → differential

2. p. 2, l. 20:
An inventory many of → An inventory of many of

3. p. 7, l. 18:
Antarctic (AO) → Antarctic (AAO)

4. p. 11, l. 21:
statistical significant → statistically significant

5. p. 16, l. 30:
our sample our located → our sample are located

6. Caption Figure 3:
with its contribution lognormal distributions → with its contributing lognormal distributions

**Suggestion:**

The current work is limited to the geographical sampling of the GPS stations. I suggest to perform (outside the scope of this paper) a similar multiple linear regression analysis for global time series (e.g. from ERA Interim and/or GOMESCIA). This could help to identify reasons for IWV variations on a regional scale.

---

## Referee Comment (RC3) · Anonymous Referee #3 · 6 Feb 2019

Overall, this is an interesting topic, but I found it hard to figure out what the paper is trying to accomplish. The purpose of the paper needs to be more clearly discussed in the first few pages. I find that the comparison between the GPS, GOMESCIA and re-analysis IPW data valuable and fairly well described in the paper. I would like to see a clearer discussion of the effect of clear-sky bias on the GOMESCIA data, and whether this could be an explanation for its larger discrepancy from the other data sources. The sorting of histograms of IPW at different sites is also valuable and clearly extends the earlier work in this area. I would like to see a more organized discussion of the impact

of seasonal behavior on these histograms. The paper asserts that the seasonal behavior is important, but does not explicitly show that if the seasonal cycle is removed, the resulting distributions of the residuals are simpler (e.g gaussian or lof-normal). I am less convinced by the "step-wise multiple linear regression." First, I don't understand the name – what is step-wise about it? Second, so many potential explanatory variables are used (which are said to be at least partly independent), that at least some of them are likely to have explanatory "power" by chance for the relatively short times series studied. I would be happier if the authors could clearly state a hypothesis, and then test it with a more limited set of explanatory variables consistent with the hypothesis. I am not a fan of a "throw everything at it and see what sticks" approach. There are numerous cases of strange English usage/wording, some of which I mention below, but there are far more than I can explicitly call out. I recommend that the paper be edited by a native English speaker. Some more detailed comments below: Page 3, line 20-22. Strange Wording ("vastly"). Also, what does neutral mean in this context? Page 3, line 30. Strange wording ("disposed of"). Maybe change to "This process results in a world-wide…." Page 4, line 29. "downsized"? How was the conversion from 5-minute observations to 6 hourly observations performed? And why is this needed/appropriate considering that the reanalysis can be considered a snapshot at the synoptic times? (I'm not saying that the downsizing is wrong, I just want it explained better. Page 8. I can't follow the discussion of the clear sky/cloudy biases. Is the GOMESCIA only available in clear-sky conditions? If so, wouldn't it be expected that the GOMESCIA is biased low compared to measurements available in all-sky conditions? Page 9 and 10. Discussion of different distribution classes at different locations. It is unclear how the sorting into classes was performed. Are all types of fits tried, and then some criterion applied? Please explain more clearly. Also, if the various distributions mostly occur because of seasonal cycles, maybe it would be good to show analysis after the seasonal cycle is removed?? Page 11, lines 19-20. This could be tested by subsetting the reanalysis data so that it too has gaps at the same time as the GPS data. Page 16. I wonder if the reason for the poor fitting for the west coast of North America sites it

due to the fact that for much of this region, the rainy season is in the winter, where the temperatures are low. In the summer, it is dry but warmer, so there is not that much of a change in IPW. Other regions, such as eastern NA or Europe, there is still significant rainfall in the summer season, leading to a much larger correlation with surface temperature.

---

## Author Comment (AC1) · 20 Mar 2019

Review of the manuscript
"Interpreting the time variability of world-wide GPS and GOME/SCIAMACHY integrated water vapour retrievals, using reanalyses as auxiliary tools" by Roeland Van Malderen, Eric Pottiaux, Gintautas Stankunavicius, Steffen Beirle, Thomas Wagner, Hugues Brenot and Carine Bruyninx

The manuscript present the results of a study focussing on the variability of integrated water vapour across the globe as provided by two different datasets (sub-daily GPS and monthly-mean GOMESCIA estimates).

I find the topic important and the results really quite interesting, and I liked the summary Figure 10. My main concern is about the way the scientific questions are addresses (too vague), the motivation behind the approaches proposed (not discussed), a lack of precision and explanations in numerous parts of the manuscript as well as too many speculations. As a result, this is sometimes confusing, so that I recommend at least a major review.

We appreciate that you find our topic important and the results interesting and we thank you for your exhaustive feedback! We agree with you and the third reviewer that the aim of the paper should be more clearly discussed in the introduction of the paper. It was provided only fragmentary at the beginning of the different sections. Also the motivation for the multiple linear regression approach (as opposed to using regional climate models which are validated by GPS retrievals, see our forthcoming study Berckmans et al. ACPD, 2018) to study the IWV time variability is mentioned at the beginning of the section, but not in the introduction.
We now clearly stated the two main research questions of the paper in the introduction:
"In this paper, we focus on answering the following questions.
1) How well are the spatial and temporal IWV variability represented by three different, independent, IWV datasets? As our primary dataset used here is a ground-based GPS IWV dataset covering a world-wide sample of 118 sites, we will assess the spatial variability only between those sites and the global IWV datasets will be sampled to the site locations. In this work, a first characterization of the spatial IWV variability is given by considering the geographical distribution of the IWV frequency distributions, an extension of the work by Foster et al. (2006). To assess the temporal IWV variability of the three datasets, we consider different time scales here: from the seasonal cycle to short inter-annual variability, and to trends over periods of around 15 years.
2) Can the spatial and temporal (inter-annual and trends) IWV variability be explained by changes of local meteorological variables (like e.g. surface temperature and pressure) and/or by low-frequency variability in atmosphere and ocean (on both global and regional scales), and if so, how? There are already a number of studies mentioning IWV patterns that result from interactions between the atmospheric circulation and the land and ocean surfaces (see e.g. Trenberth et al., 2005, Wagner et al., 2006, Shi et al., 2018, Wang et al., 2018). These studies are based on correlation studies between IWV and one such a teleconnection index (ENSO: all mentioned studies, Pacific Decadal Oscillation: see Shi et al., 2018). Here, we fit the monthly mean IWV time series by means of a stepwise multiple linear regression approach with  proxies for the seasonal cycle and linear trend, and with local meteorological variables and teleconnection indices as explanatory variables. With this empirical approach, we aim at finding out which the most relevant variables are to explain the IWV variability for different regions, independent of the used IWV dataset. To our knowledge, it is the first time that such an analysis is done on the IWV time series of individual sites."

For your general comment on a lack of precision and explanations in numerous parts of the manuscript, we went through the manuscript again and added explanations to the issues raised by you, but also identified by ourselves. About the high number of speculations: we know that especially Sect. 6.2.3 (with its summary graph Fig. 11) is speculative, because this is the first time that so many teleconnection indices are linked, all together, with the IWV in such an empirical approach, so we could not refer to existing literature.

Some examples of lack of precision:
1) the teleconnection indexes are presented with no logical link, for instance, you do not explain your choice, nor systematically provide the associated fluctuation time scales; as a result this is confusing;
The selection of the teleconnection indices is done by analyzing ourselves their correlation with IWV for different regions (see Sect. 6.2.1.). We made a reference to this subsection here.
The complete description of the explanatory variables is out of scope here, but we reorganized Sect. 2.4 and give examples how they differ (atmospheric vs. oceanic circulation, spatial scale of impact, time scale of their variability).

2) The section describing the GPS dataset is quite long but I could not find information on how the authors dealt with missing data;
As we consider the GPS dataset as our the primary IWV dataset here, and as this manuscript has been submitted to the ACP/AMT/ANGEO Special Issue "Advanced Global Navigation Satellite Systems tropospheric products for monitoring severe weather events and climate (GNSS4SWEC)", we think that a long description of the GPS IWV retrieval is needed here. We do not deal with the missing data: for the

analyses here (seasonal cycle, linear trend, multiple linear regression), they are not at all problematic, although they had an influence on the lower correlation coefficients for GPS when fitting harmonic functions to determine the phase and amplitude of the seasonal cycle. This has been added in the manuscript, as a response to a suggestion by reviewer 3. In the manuscript, after mentioning the percentage of gaps in the data, we added that we did not specifically deal with missing data here.

3) in the regression analysis, I am not sure surface pressure and atmospheric temperature are taken at the site location or over the region to which the site belongs;
The surface pressure and surface temperature are calculated from the respective values from the four grid points surrounding the site by horizontal interpolation, weighted with the inverse distance to the site. To account for the height difference between the site location and the ERA-Interim surface grid points, we assume a standard temperature lapse rate of −6.5 K km−1 (typical for wet adiabatic conditions) for the surface temperature altitude correction, and the hydrostatic and ideal gas equations to adjust the surface pressure. A reference to this text, in Sect. 2.1 and Sect. 2.3.1 has been made when the meteorological parameters are first mentioned as candidate explanatory variables in the multiple linear regression.

4) sentences such as "for (sub)tropical sites and sites in East Asia two distinct lognormal distributions are needed, probably related to the monsoon and ENSO" why ENSO?
We added the reference to Foster et al. (2006) here, in which they showed the frequency distributions of some of those sites for different years. The authors could determine links between the shape of the frequency distribution of that year and the fact that the year was an El Niño, La Niña, or "normal" year. We also investigated this for our sample of sites and came to similar results. However, because of the interest of space, we decided not to include this in our manuscript.

5) in Figure 4, there are more tones of colours than indicated in the figure caption (e.g. pale green versus darker green, same for yellow to green).
This is done by purpose. The colour scale in the figure is continuous, while the colour bar is discrete. Here below, we add the same figure, but now with the colours used in the dots corresponding one to one to the colours of the colour bar. To our opinion, with this approach, we cannot represent enough the subtle gradients in GSD within a geographical region. Therefore, we propose our approach and added the following explanation to the figure caption: "Please note that the colour bar is a discrete indication of the colouring for the specified ranges. The colouring of the dots is done by a continuous scale, to better highlight the subtle GSD differences within a region".
If you disagree with us, we can of course replace the figure, if requested.
The same representation by a continuous colour scale has been adopted in the (old) figures 7, 9, 10.

[Figure]

6) "Comparing our mean IWV trends with the 0.26 mm decade-1 GPS IWV trend quoted by Wang et al. (2016a), we found slightly lower rates of 0.19, 0.08 and 0.11 mm decade-1", 0.10 is NOT slightly lower than 0.26!
We removed "slightly".

It may be a good idea to separate the manuscript in at least two parts, so that it may be easier to present proper presentations and discussions of the results (which are very interesting and numerous). Including more material in appendix may help too.
We thank you for this suggestion, but we are not in favour of splitting the manuscript in two parts. First of all, it is not a short paper, but also not excessively long. Secondly, the paper deals with the analysis of the

properties and spatial/time variability of the IWV measured at GPS sites, starting from its most simple statistical representation (the frequency distribution), over the seasonal cycle and linear trends to the inter-annual variability. Therefore, we consider this study as whole and prefer not to split it. We also do not see a solution how to do it. If you have some ideas about it, we would be happy to receive them.

Also, I think it would be excellent to take advantage of the global dataset (GOMESCIA) to add maps covering the whole globe rather than only presenting maps with results shown at GPS sites (data are sparse in large areas of the world and very clustered in others).
Here too, we do not completely agree with you. This paper is submitted to the ACP/AMT/ANGEO Special Issue "Advanced Global Navigation Satellite Systems tropospheric products for monitoring severe weather events and climate (GNSS4SWEC)", so the focus of this study are the GPS IWV retrievals at the sites. Therefore, we only considered the GOMESCIA and ERA-Interim IWV time series sampled at those sites. If we would show global maps based on GOMESCIA and ERA-Interim, and plot the GPS results on top of it, this would give the impression that GPS is used as a validation dataset for the other two datasets, which is exactly the opposite of our aim here. Moreover, some of the maps for ERA-Interim and GPS are already available in Parracho et al., ACP, 2018 and a similar study is under development for GOMESCIA, for which the first results have been presented at the EGU 2018: https://meetingorganizer.copernicus.org/EGU2018/EGU2018-9217.pdf. Also the second reviewer agreed that such an analysis is outside the scope of this paper. We nevertheless included the suggestion in the outlook.

On another subject, while I can understand your choice for a regression method where you test very numerous potentially explaining variables, I think that you should discuss more how your results compare or not with other studies and mechanism (I understand that it is difficult because there is a lot in the manuscript).
If the same explanatory variables were consistently retained for sites belonging to different regions, we tried to explain or discuss the relevant mechanisms behind this correlation with IWV. Unfortunately, as written in the introduction of our manuscript, at present, there are no other studies performing a multiple linear regression for IWV with those explanatory variables. The few that we mentioned performed a correlation analysis between one explanatory variable and IWV. Most of the past analyses try to link the explanatory variables with surface temperatures and precipitation, but not with IWV. We know that therefore, some of the arguments might sound hypothetical or too vague, but there is simply not much literature up to now.

Also, I did not understand your motivation for including the linear trend as an explanatory variable while it was already taken into account in Eqn (1) and finally did not explain much more.
We apologize for being not clearer on this point, also the second reviewer addressed the same issue. **All the variables (describing the seasonal cycle, the linear trend and the explanatory variables) appearing in Eq. (1) are included only in the multiple linear regression if they significantly contribute to the regression coefficient. So, in most of the sites, the linear trend is not included in the final multiple linear regression. We hope that this has been clarified in the manuscript now.**

Below I present some specific comments on the first part of the manuscript. This is not exhaustive, but I hope they can help the review process and help you to revise the following sections and conclusions accordingly for a second round of review.

Specific comments
Abstract: you need to be more precise about the time periods and time scales of analysis.
Page 1, line 14, "IWV variability": please precise at which time scale and over which period, and the IWV sampling time step.
We added the period and for the different analyses we did, we mentioned the IWV sampling time step.
Page 1, line 18, "on average": this is too vague, on average over what?
Averaged over all stations, this is added to the text.
Page 1, line 20-21, "the seasonal behaviour and the long-term variability are fitted together": this is not exactly what I understood. Rather, you aim at reconstructing the time series of monthly-mean IWV from the mean annual cycle, linear trend and explanatory variables.
We changed the text to "Finally, we reconstruct the monthly mean IWV time series by means of a stepwise multiple linear regression from the mean annual cycle, the linear trend, and a selection of regionally dependent candidate explanatory variables."
Page 1, line 25, "long term trend": please precise, i.e. linear trend over the period [year1,year2]
Done.
Page 1, line 26: variableS
Changed.

Introduction

In the last paragraph of the introduction, you present the work presented but I could not really find a clear presentation of the question(s) you want to address. In my opinion, an analysis of a new type is not a good enough motivation per se.

We provided a detailed description of the research questions and how we handled them in the manuscript (see above).

Page 2, line 4, 'on local scale': are you referring to mesoscale here? I would rather say that "At all scales" rather than local scale because precipitation for instance is not simply related to "local water vapour" alone; it typically involves larger, e.g. synoptic atmospheric circulations as well. In addition to the diabatic processes you mention, I would add radiative processes. I would remove "Of course".

Done.

Page 2, line 7, 1st sentence: a reference is needed there.

Done, we referred to Kämpfer, 2012.

Page 2, line 26, "have the potential to be used for climate change analysis, which is the subject of this paper": I do not agree about this statement, the results of this study are more focused on interannual variability (and trend over the 20-year-long period).The present study provides very valuable information about IWV variability in space and time. In time, it goes from the annual cycle to short inter-annual variability to trends over periods of 20 years or less. In my opinion, 20 years is too short to provide robust information on climate change. It is less than the time interval typically used to compute climate mean (30 years, e.g. see http://www.metlink.org/climate/depthclimate- met-office/).

We agree with you that for climatological means, time periods of 30 years are used (also at our meteorological institute). So, we are not posing that with our study, we can actually observe the climate change. But, as we know that the surface is warming, with this study, we would like to investigate the effect of e.g. this warming on the water vapour variability, at the time scales mentioned by you. We changed this statement in the text to. "In such a study (Van Malderen et al., 2014), we compared five different techniques and concluded that the techniques used here (GPS and GOMESCIA) are promising to study the inter-annual IWV variability within the context of a warming climate, which is one of the aims of this paper."

Page 2, line 31: add 'generally' before 'occur' as this is to my knowledge not strictly true for all geographic locations.

Done.

Page 3, line 2: remove 'can' as you precisely provides number who illustrate it.

Done.

Page 3, line 4: remove "of course" and replace "can be" by "are" as existing studies allow you to be more affirmative.

Done.

Page 3, line 6: I am not sure of what you mean by "autocorrelation" here.

Page 3, line 6, about ENSO: it seems to me that, more precisely, the relatively large magnitude of the signal induced by ENSO events at inter-annual scale affects trends computed on periods of 10-20 years. It would be good to reformulated a bit the sentence to be more informative.

We agree with you that this sentence does not add significantly new information with respect to the previous one. We therefore decided to remove it.

Page 3, lines 16-17: I think it is "on one hand", not "at one hand". More generally would be useful to check English throughout the manuscript. Avoid expressions such as "not surprisingly" or "of course" when you do not provide explanation nor references.

We went through the entire manuscript again and tried to improve the English and avoid using too suggestive language.

Page 4, lines 7-14, about homogenization: as the dataset in use here is not homogenized, I think this paragraph is unnecessary long. You could mention in the conclusion "for extension/improvement of this study, the use of a future homogenized dataset (as described in Van Malderen et al. (2017)".

We shortened this paragraph, but we still think it is necessary to point out that inhomogeneities can have a large impact on the calculated trends. We also made reference to this activity in our "Conclusions and outlook" section.

Page 4, presentation of GPS data processing: you may consider moving part of it in supplementary material.

As we already mentioned, as this manuscript has been submitted to a Special Issue with focus on GPS, we feel obliged to spend quite some text on the data processing, so that interested readers could compare this study with other published studies based on the same IGS repro 1 datasets (Wang et al., 2016, Parracho et al., 2018). Moreover, in this section, we also describe how the surface temperature and surface pressure time series (used in Sect. 6) were sampled and corrected to the GPS site locations and altitudes.

Page 6, about ERA5: this is not used, so the whole sentence is useless. It could be used in the future only, so potentially, you could mention it in the perspectives.

We removed this sentence and made a reference to ERA5 in the "Conclusions and outlook" section.

Page 6, lines 16-22: I am wondering why you mention this with so much details without linking this to your study.

We changed this paragraph to "The homogeneity of the extracted ERA-Interim IWV time series has been questioned recently by Schröder et al. (2016) and Ning et al. (2016) due to changes in the observing systems or changes of the input to assimilation schemes."

Page 6, line 32: it seems to me that prior to Chen and Liu (2016), other studies such as by Bock et al. already extensively evaluated ECMWF and NCEP products.

Yes, but we only quoted the most recent one here.

Page 7, presentation of teleconnection indices: these indices are presented without much logic, you must re-write this section in a way that motivates your choice, and explain more over which time scales/regions they are relevant (and add somewhere the precise coordinates of the regions presented in the last figure).

The selection of the teleconnection indices is done by analyzing ourselves their correlation with IWV for different regions (see Sect. 6.2.1.). We made a reference to this subsection here.

The complete description of the explanatory variables is out of scope here, but we give examples how they differ (atmospheric vs. oceanic circulation, spatial scale of impact, time scale of their variability).

Page 8, section 3: 1) how did you deal with missing GPS data? 2) It is not well suited to use the word 'bias' as you do not have a reference dataset here.

We added "we also computed the statistical parameters mentioned here below only for the months for which the GPS monthly mean IWV dataset has actually values"

Strictly statistically speaking, you are right about the use of the word bias. But, this is a commonly used term when comparing datasets for a variable (including IWV, see http://www.meteo.be/IWVintercomp), without a clear reference dataset. For humidity/water vapour measurements, we speak about a dry bias (negative mean difference) and wet bias (positive mean difference) of one technique to another (not necessarily a reference).

Page 8, "We exclude the GOMESCIA dataset here, as only monthly means are available, which might be problematic to compute significant frequency distributions": I do not understand what you mean. It seems you are mixing statistical robustness and time scale issues.

With only monthly means, you have at most 186 data points in your frequency distribution, which is not enough to be fitted well by lognormal or Gaussian statistical frequency distributions. We tried it, for the IWV monthly mean time series of the 3 datasets. You need either longer time series or capture the intra-monthly IWV variability (as we did with the 6h time sampling) to have observational frequency distributions that could be fitted well. So, the exclusion criterion is purely technical (or statistical robustness) and the motivation is not to study the intra-monthly IWV variability in particular. We clarified this in the text: "We exclude the GOMESCIA dataset here, as only monthly means are available, which puts constraints on the statistical robustness of the observational frequency distribution and its fit by a (log)normal distribution".

Pages 8, 9, 10 and Figure 4: there are more tones of colours than indicated in the figure caption (e.g. pale green versus darker green, same for yellow to green). Please clarify.

See our answer on your general comment 5) on Figure 4.

Also, I would like to see the full maps obtained with ERA-I, NCEP as well as monthly-mean GOMESCIA IWV values. This would allow assessing the representativeness of the results obtained at the sites, and provide a clearer picture than emerging now at the end of this section.

We again refer to our response to your general comment about constructing those full maps based on the gridded datasets like GOMESCIA, ERA-Interim and NCEP/NCAR. We agree that this approach will provide an overall clearer picture, but we also agree with the second reviewer that this is out of the scope of this manuscript and the Special Issue it is submitted to. Moreover, this is also the subject of forthcoming work. We added this to the "Conclusions and outlook" section.

It is also very difficult to see the results over Europe because circles are overlapping each other.

Yes, we know. What do you suggest? Adding a snapshot over Europe for all figures to the Supplementary material?

Page 11, Figures 5 and 6: A discussion of the geographical patterns shown in Figure 5 is missing.

We added a small discussion on the geographical patterns shown: "This graph shows that the IWV seasonal cycle peaks in the summer months at both hemispheres. The amplitude of the seasonal cycle is largest for the Asian Northern Hemisphere sites, where there is a distinct difference between a dry and wet season. In Northern America, the central and eastern sites have larger amplitudes than the western coast sites, which is related to the fact that for most of these latter sites, the rainy season is in the winter, when the temperatures are low, while the summer are dry but warmer. In Europe, the Mediterranean sites have smaller amplitudes and the IWV seasonal cycle peaks one month later compared to the rest of the continent."

We do not want to be too detailed here, as we focus here on how well the different datasets compare at representing the seasonal cycle at our sample of stations (see also next comment).

In Figure 6, I am wondering about how this graph was made: did you consider all the stations? I guess the uneven location of the stations is playing a large role in the shape of these histograms. I would like to see the same graph with all the global GOMESCIA dataset.
Yes, we considered all the stations. We specified this in the text. The uneven location of the stations is indeed playing a large role in the shape of the histograms of the amplitude and phase. However, the precise shape of those histograms does not matter here, as we focus on how well the different datasets compare in representing the seasonal cycle at our sample of stations (we refer here to the first scientific question that is addressed by this study), and are not interested in the global frequency distribution of the seasonal cycle phase and amplitude an sich. Therefore, adding the same graph making use of the global GOMESCIA (and ERA-Interim) datasets is not relevant here.

Page 11, section 6: Given the content of this section, I suggest that you modify the title (especially remove "long-term"). You could rather emphasize the idea of the trend over [year 1, year 2] and inter-annual variability in this time window. You could probably shorten your discussion of the statistics, and it would be clearer if you could add a few words about decadal and multi-decadal variability.
The title has been changed. Given the fact that only 1.5 decade is available for the datasets used here, we do not see which meaningful discussion could be added here about decadal and multi-decadal variability. Please share with us your ideas about this.

Page 11, lines 2-3: "As we have only 15 years available for most of the stations, our time series is too short to draw firm conclusions on the presence or magnitude of a trend." Then, just below you compute trends. The way it is written is very confusing. I suspect you mean an "expected climatic trend" in the first instance.
The point that we want to make here is that we will not discuss the magnitude or the presence of a trend (they cannot be statistically significant according to the formalism of Weatherhead), but we want to focus on the differences of the trends between the different datasets and on the interpretation of the inter-annual variability. So, we will not make firm statements about "IWV is increasing at this region", but merely "at this region, the three different datasets compare well and show an IWV increase". We tried to make this clearer in the manuscript.

Figure 8: It is important to add the residual obtained when removing the mean annual cycle and the linear trend to the time series, in order to be able compare it with its magnitude with the magnitude of the residual that you show.
The second reviewer also did not understand very well what was in this figure. For the two stations in this figure, the linear trend was not included in the multiple linear regression, as it did not contribute significantly enough to the correlation coefficient. The mean annual cycle was included for both sites. As requested by the second reviewer, we added some intermediate steps for the multiple linear regression procedure for those 2 sites as extra figures in the Supplementary material. We hope that this helps in the interpretation of this figure. We also added some extra clarifications in the figure caption.

I have more specific comments to come on a revised version of the manuscript where you would have taken into account my general and specific comments, as I think this manuscript should be published (perhaps as a two-part paper).

---

## Author Comment (AC2) · 20 Mar 2019

In this paper an analysis of integrated water vapour (IWV) data from three different sources (ground based GPS data, GOME/SCIAMACHY/GOME-2 satellite data and ERA Interim model data) is presented. Beside intercomparisons of the three data sets major topics of the paper are the analysis of time series and the association of variabilities in IWV with 'explanatory variables', e.g. physical / dynamical processes described by teleconnection indices.

This is an interesting approach which is well presented, but some aspects need further clarifications. Especially, the method of stepwise linear regression needs to be explained in more detail. Which terms of eq. (1) are fitted in the individual steps (and in which sequence), which corrections are applied to the data sets used in each step? Especially, it should be clarified if/how e.g. seasonal variability and trends of the 'explanatory variables' is considered. Have these data been de-seasonalised before the fit? If not, how are correlations with the linear trend and seasonal terms in eq. (1) handled? It seems that e.g. linear trends are sometimes not fitted at all because they are covered by explanatory variables variations - how is this decided? Does the sequence of fitted variables matter?
Thank you very much for your positive feedback. We elaborated more on the description of the method and the practical issues that you mentioned here in the manuscript. You will also find our answers to these questions here below, when they were repeated in your specific comments!

Maybe it would help to add e.g. for one example a plot of the different fit steps with the different time series used in each step (not necessarily for all explanatory variables, but for the major ones); this could possibly be an extension of Fig. 8. Another aspect is the selection of relevant explanatory variables for the fit. I would assume that many of the variables (e.g. temperature, pressure, precipitation) are strongly correlated - it needs to be explained in more detail how this is considered in the fitting procedure as well as in the interpretation of the results.
We did not change Fig. 8 (now Fig. 9), but instead added two extension figures of Fig. 8 in the Supplementary Material: in one figure (S2), we present the time series of the 5 most important explanatory variables that have been included in the multiple linear regression fit of both the good and bad example. In a subsequent figure S3, we then show how the fit evolves when adding each of those explanatory variables, step by step, to the multiple linear regression.
The other part of your comment (about the correlation of the explanatory variables) is treated in our response to your specific comment 7 and is also elucidated in the manuscript.

**Specific Comments:**
1. p. 6, l. 28–29:
'The database has been enhanced with many observations not available in real time for operational use.'
What is meant with this? Which database/observations do you refer to? Did you use/produce a dedicated version of NCEP/NCAR data? Please clarify.
It is meant here that the data assimilation database is enhanced with many observations that have not been used in the operational version of January 1995 that was taken to perform the reanalysis. We adapted this sentence to "The assimilation database has been enhanced with many observations not available in real time for operational use at that time."

2. p. 9, 1st paragraph:
As I understand, the difference between lognormal and reversed lognormal distribution is that there is either a (non-zero) lower or upper limit for the IWV. Is there a physical reason why a lognormal distribution occurs for subtropical and temperate climate and a reversed lognormal form in tropical oceanic environments?
This is indeed a very interesting question. Foster et al. (2006) also reflected on this issue. The absolute lower limit (threshold parameter t=0) is simply a completely dehydrated atmosphere, while the most obvious interpretation for the upper limit in the reversed lognormal situation is that it corresponds to some maximum carrying capacity of the atmosphere: complete saturation of the atmospheric column up to some characteristic level for example. So, reverse lognormal distributions appear to be connected with areas that experience almost total saturation, typically oceanic equatorial zones, while for subtropical and temperate climate zones, the lower bound is more decisive. A possible interpretation of a simple lognormal form for a IWV time series is that the source region(s) for the observed moisture is(are) effectively mixed over the considered time period, as this ensures that one mean and variance can be used to describe the region(s). This explanation is shortly added to the text.

3. p. 9, l. 10ff. and Fig. 3:
Since the 'shouldered' lognormal distribution is a new category I suggest to add also an example plot for this in Fig. 3.
Fig 3.c already shows an example of this category. This is also clarified in the text.

4. p. 11, l. 5–6:
'the IWV seasonal cycle for about 15 sites in the Northern Hemisphere peaks one month later in the GOMESCIA dataset with respect to the GPS and ERA-Interim datasets' Is there an explanation for this?

The GOMESCIA IWV peaks for these sites more frequently in August than in July, compared to GPS and ERA-Interim, but the difference between the mean July and August values are often very small at those sites. We refined this in the text. Part of the difference can be ascribed to the diurnal variation of the IWV and the difference between the fixed satellite overpass times for GOMESCIA and the 6h time sampling for the GPS and ERA-Interim. For about 10% of the sites, we find a difference of one month in the IWV peak between the monthly means calculated e.g. at 0h and 12h UTC for GPS and ERA-Interim. The clear sky observation bias for GOMESCIA might contribute as well.

5. p. 12, l. 7–8:
'We calculated linear trends as the slope of the linear regression line that was fitted (by minimising the least squares) through the monthly anomaly IWV time series.'
Please explain how these anomalies are calculated (e.g. by subtraction of harmonics or long-term monthly means).

We added the sentence "obtained by subtracting the long-term monthly means from the monthly averages"

6. p. 12, l. 16–18:
Is here an explanation for the differences in GPS trends from this study and Wang et al. (2016a)? If the same data set is used (as stated in section 2.1), why are there differences?

There are different explanations. First of all, we used the same raw data set (IGS network of 117 (Wang et al. 2016a)/118 sites with data extending from 1995/1996 until 2010/2011), but the ZTD processing might be different (IGS repro 1 in our case, not clear which processing has been used for the different periods by Wang et al. (2016a). The conversion from ZTD to IWV has been done in our study from the ERA-Interim surface pressure and weighted mean temperature at the 6h temporal grid. For constructing the 2-hourly IWV dataset in Wang et al. (2016a), Ps is derived from global, 3-hourly surface synoptic observations with temporal, vertical and horizontal adjustments, and Tm is calculated from NCEP/NCAR reanalysis with temporal, vertical and horizontal interpolations. And finally, the trends calculated in our study range from 1996-2010, while in Wang et al. (2016a), the time period 1995-2011 is considered. In our manuscript, we included "The raw GPS dataset used in their study and ours is identical, but the trend difference can be explained by different factors: a different data processing, the use of different meteorological data sources for the ZTD to IWV conversion (see the Appendix for e.g. an assessment of this factor on the trends), a different time resolution (and temporal interpolation in Wang et al., 2016a), and the use of a different time period for which the trends were calculated (see above)".

7. p. 14, 1st and 2nd paragraph:
Are seasonal variations considered in the checks for independency of explanatory variables? Where does the limit of 0.90 for the linear correlation coefficients come from (in the section on linear trends a correlation of $R_2$=0.66 is considered as large).

Yes, we calculated the correlation between the entire time series of the (global) explanatory variables (including the seasonality) like the teleconnection patterns. But additionally, for every site separately, we calculated the correlations between the local meteorological parameters (surface pressure, surface temperature, tropopause pressure, precipitation), and between these local meteorological parameters and the teleconnection patterns. The 0.90 limit for the linear correlation coefficient is chosen so high to exclude only strongly correlated explanatory variables at this stage. This has also been clarified in the manuscript. We are confident enough in our statistical t-test in the stepwise multiple linear regression technique not to retain an explanatory variable if, in the previous step, another explanatory variable with which it is strongly correlated, was already selected. See also immediately here below for more explanations.

In this context, and also for the regression tests used to determine the significance of the different explanatory variables: Have the data sets been deseasonalised before the comparisons? Is the preparation of variables for this test consistent with their later use in the fit (see also general comment)? Is the fact that 'Variables with a significance level lower than 5% are discarded' the reason that in the fits instead of 100-200 variables only 6-8 (see end of this page) are considered?
Please explain.

We did not make any corrections or did not deseasonalise the time series of the explanatory variables for the linear regression. The time step of the linear regression is one month, so we used the monthly mean time series for both the IWV and all explanatory variables. Some of the teleconnection patterns are provided as monthly means on the web pages mentioned in Table S2, for the other explanatory variables, we calculated monthly means based on daily values or on the 6h time grid (surface temperature, surface pressure, tropopause pressure).

8. p. 14, l. 17–19:
'a significant positive trend is still present in the residual time series (although the annual trend was not retained as a significant explanatory variable in the multiple linear regression).' This is unclear. If (as in eq. (1)) the linear trend is fitted, why is there a remaining trend in the residual?

The linear trend is considered as an explanatory variable as well in the stepwise multiple linear regression, and is included only if it significantly contributes to the regression coefficient (by means of a t-test, at the 95% significance level). We clarified this in the text.

9. p. 14, l. 23–25:
'part of the seasonal behaviour present in the time series still has to be explained by other variables, especially by the surface temperature and precipitation time series'
As mentioned in the general comments, it has to be clarified how seasonal variations are considered in each of the fitted time series and how this is in line with the seasonal terms in eq. (1).

The term(s) describing the seasonal behavior in Eq. (1) are treated as explanatory variable(s) as well, just like the linear term, and they are included only in the multiple linear regression if they significantly contribute to the regression coefficient. We hope this is now also clarified in the manuscript itself.

10. p. 17, l. 23–25:
'the NAO index is present in only one third of the sites as explanatory variable, although its relationship with precipitation is well established in Europe' If NAO index and precipitation are closely related, why can these be considered as independent explanatory variables?

As shown in the referenced paper, the NAO index and mean/extreme precipitation are spatially correlated, especially in summer and winter, and with different signs in different parts of Europe. This does not necessarily mean that their time series are also strongly correlated. For every European site we considered here, the linear correlation between the precipitation and the NAO index never exceeds 0.5 (in absolute terms).

11. p. 18, l. 26:
'we do not expect an effect of IWV on precipitation' Why not? Please explain.
We removed this sentence.

12. p. 19, l. 10–13:
'Moreover, whereas the majority of the sites have positive trends in their IWV time series, especially for the GPS and ERA-Interim datasets (see Sect. 6.1), the residual time series after applying the multiple linear regression show an equal amount of positive and negative trends (GPS and ERA-Interim) or even a higher amount of sites with a negative trend (GOMESCIA).'
Obviously, there are different ways to determine trends. As I understand, trends from section 6.1 originate from a simple linear regression (and probably a to be defined seasonal correction, see above). It seems that the multiple linear regression trends mentioned here are those remaining after subtracting the effects of explanatory variables without a linear trend fit (although a linear term is given in eq. (1)). This should be clarified.

As should be explained now here before and in the manuscript, the residual time series is obtained by subtracting from the original IWV monthly mean time series the multiple linear regression fit in eq. (1), which might contain the long-term monthly means, a linear function and explanatory variables, depending on their significant contribution to the regression coefficient. We also clarified this in the text. The main message that we want to give here is that some of the explanatory variables, besides the linear trend term, are able to explain the linear trends discussed in Sect. 6.1.

13. p. 19, l. 25–27:
'the linear trend sign of the explanatory variable's term (coefficient multiplied with its time series) is in agreement with the linear trend sign of the IWV time series of the same site'
It is not clear what is meant with 'trend sign of the explanatory variable's term' and how this is derived. Do you fit a linear trend to the explanatory variable's term? Please explain.

We give an example here. Suppose that an IWV time series shows a positive trend (as calculated in Sect. 6.1). For this time series, among others, e.g. the surface pressure was kept as an explanatory variable. This is time series $X_j(t)$ (with e.g. j=2) in Eq. (1). At this station, the surface pressure decreased over the time period considered. So, for this station, the linear trend in IWV could not be explained by the linear trend in the surface pressure, unless the coefficient $B_j$ (with j=2) is negative. Let us assume it is indeed negative. So, for this example, the linear trend sign of the surface pressure term $B_jX_j(t)$ (with j=2) (positive) is in agreement with the linear trend sign of the IWV time series (positive). We are aware of the complexity of the formulation used here and we propose to change into: "As the time series of the local meteorological variables Xj(t) appear with coefficients Bj in the multiple linear regression equation (1), a positive trend in Xj(t) might be compensated by a negative coefficient Bj, so that the negative trend in BjXj(t) could still be linked to an IWV decrease at that site, for instance. As a matter of fact, more or less independent of the

dataset used, we found that, on average, for about 70% of the cases for which the mentioned (local) explanatory variables Xj(t) are present in the multiple linear regression of the sites, the (linear) trend sign of the explanatory variable's term BjXj(t) is in agreement with the (linear) trend sign of the IWV time series (see Sect. 6.1) of the same site."

14. p. 20, l. 25–27:
'the precise identification of the main contributor to the IWV trend is almost impossible'
This sounds rather pessimistic. In most cases there will not be one single contributor to the IWV trend. The analysis presented in this paper at least shows for certain regions the main contributors, and this is a very useful result which could possibly even be the basis for further investigations (see suggestion below).
Thanks for this more optimistic formulation. We changed it in the text as well.

15. p. 22, l. 14:
The 'meteorological station' is only introduced in the next paragraph, should be explained before.
Solved by using the more general term "data source".

16. p. 22, l. 14:
'the slope of the linear regression (with correlation coefficient 0.84) between the Ps and IWV biases between the different corresponding datasets for the 40 IGS stations is equal to the -0.34, confirming the acceptable data quality of the pressure observations at the retained stations.'
It is not fully clear to me what has been done here and how the derived numbers are to be interpreted. Do you refer here to the begin of the appendix 'a 1 hPa change in Ps gives an IWV change of 0.36 mm'? Table A1/A2 list three different Ps sources - to which do you refer here?
Yes, we refer here to the 'a 1 hPa change in Ps gives an IWV change of 0.36 mm'. As is made clearer in the text now, the correlation is made between "the individual values of the first column of b), not the mean".

17. Table A1:
(a) Please define "abs bias" and "abs trend". Probably these are the bias and trend of the absolute differences? Why are these absolute numbers relevant in relation to the interpretation of the multilinear fit?
We changed this to "In Table A1, we present the means – weighted by the number of observations for each station – of (from left to right) the IWV differences, the absolute value of the IWV differences, the standard deviation of the IWV differences, the linear correlation coefficients between the IWVs, the IWV trend differences, and finally the absolute IWV trend differences between two GPS IWV datasets that disagree only by one or more of these auxiliary meteorological parameters."
This appendix is not directly relevant for the interpretation of the multilinear fit, but should give an idea how sensitive the linear IWV trends calculated in Sect. 6.1 are on the used meteorological parameters needed to convert the GPS ZTD to IWV.
(b) What exactly is meant with case a)? Is this a comparison between the IGS and the ERA Interim IVW data?
Indeed! So, this is an addition to the Sect. 3 "Dataset comparison" for ERA-Interim and IGS. We added this reference to this section in the text.

(c) What is meant with 'the two different databases of the meteorological variable whose impact on the IWV is studied'? For example, I interpret case b) as a comparison between IWV results based on Ps from ERA Interim and Ps from synoptic stations (this should be the non-italic numbers). What are the italic numbers in this case?
You are right about your interpretation of the plain numbers for case b). We wrote in the manuscript, immediately after the cited text to answer your issue 17 (a): "The numbers in italic are more informative and denote these same statistical means of (trend) differences, but now for the meteorological parameter that differs between the two datasets (resp. Ps, Ts, and Tm for cases [b], [c], and [d])."

18. Figure 4:
Why do you only show Classification for GPS (a) and GSD for ERA Interim data (b)? I assume these plots should be available for both data sets.
These plots are similar for the other datasets. We nevertheless included them in the Supplementary Material.

19. Figure 8: It should be clarified that no linear trends are fitted, only proxies (if this is the case here).
We hope this is clear now.

**Technical Corrections:**
1. p. 2, l. 16:
differntial → differential
Corrected.
2. p. 2, l. 20:
An inventory many of → An inventory of many of
Corrected.
3. p. 7, l. 18:
Antarctic (AO) → Antarctic (AAO)
Corrected.
4. p. 11, l. 21:
statistical significant → statistically significant
Corrected.
5. p. 16, l. 30:
our sample our located → our sample are located
Corrected.
6. Caption Figure 3:
with its contribution lognormal distributions → with its contributing lognormal distributions
Corrected.

**Suggestion:**
The current work is limited to the geographical sampling of the GPS stations. I suggest
to perform (outside the scope of this paper) a similar multiple linear regression analysis
for global time series (e.g. from ERA Interim and/or GOMESCIA). This could help to
identify reasons for IWV variations on a regional scale.

Thank you for this suggestion, we included it in the conclusions. As a matter of fact, such a study is under development and the first results have been presented at the EGU 2018: https://meetingorganizer.copernicus.org/EGU2018/EGU2018-9217.pdf

---

## Author Comment (AC3) · 20 Mar 2019

Overall, this is an interesting topic, but I found it hard to figure out what the paper is trying to accomplish. The purpose of the paper needs to be more clearly discussed in the first few pages.

The same comment has been raised by the first reviewer. We now clearly stated the two main research questions of the paper in the introduction:

"In this paper, we focus on answering the following questions.

1) How well are the spatial and temporal IWV variability represented by three different, independent, IWV datasets? As our primary dataset used here is a ground-based GPS IWV dataset covering a world-wide sample of 118 sites, we will assess the spatial variability only between those sites and the global IWV datasets will be sampled to the site locations. In this work, a first characterization of the spatial IWV variability is given by considering the geographical distribution of the IWV frequency distributions, an extension of the work by Foster et al. (2006). To assess the temporal IWV variability of the three datasets, we consider different time scales here: from the seasonal cycle to short inter-annual variability, and to trends over periods of around 15 years.

2) Can the spatial and temporal (inter-annual and trends) IWV variability be explained by changes of local meteorological variables (like e.g. surface temperature and pressure) and/or by low-frequency variability in atmosphere and ocean (on both global and regional scales), and if so, how? There are already a number of studies mentioning IWV patterns that result from interactions between the atmospheric circulation and the land and ocean surfaces (see e.g. Trenberth et al., 2005, Wagner et al., 2006, Shi et al., 2018, Wang et al., 2018). These studies are based on correlation studies between IWV and one such a teleconnection index (ENSO: all mentioned studies, Pacific Decadal Oscillation: see Shi et al., 2018). Here, we fit the monthly mean IWV time series by means of a stepwise multiple linear regression approach with proxies for the seasonal cycle and linear trend, and with local meteorological variables and teleconnection indices as explanatory variables. With this empirical approach, we aim at finding out which the most relevant variables are to explain the IWV variability for different regions, independent of the used IWV dataset. To our knowledge, it is the first time that such an analysis is done on the IWV time series of individual sites."

I find that the comparison between the GPS, GOMESCIA and reanalysis IPW data valuable and fairly well described in the paper. I would like to see a clearer discussion of the effect of clear-sky bias on the GOMESCIA data, and whether this could be an explanation for its larger discrepancy from the other data sources.

The scope of this paper is not the comparison of the different IWV datasets. This was the subject of our previous paper (Van Malderen et al., 2014) and has also been discussed in Beirle et al. (2018). However, we added some extra information from these papers in the manuscript, e.g.:

"The selection of cloud-free observations for GOMESCIA - unavoidable for water vapour retrievals from satellite measurements in the visible range, where cloudy scenes have to be masked out - corresponds to generally dryer atmospheric conditions, which likely results in low biased means. Comparisons to independent measurements in Beirle et al. (2018) result in relative biases of typically -5 to -10% for the total mean. In this context, it should be noted here that the GOMESCIA climate product was optimized for inter-instrumental consistency over time, not for accuracy (see Beirle et al., 2018). However, the impact of the dry bias on trend analysis is small, unless also the cloud properties themselves change over time."

The sorting of histograms of IPW at different sites is also valuable and clearly extends the earlier work in this area. I would like to see a more organized discussion of the impact of seasonal behavior on these histograms. The paper asserts that the seasonal behavior is important, but does not explicitly show that if the seasonal cycle is removed, the resulting distributions of the residuals are simpler (e.g gaussian or log-normal).

This suggestion has been taken into account by including a figure (new Fig. 5) of the classification after the seasonal cycle has been removed and a description of this analysis in the manuscript. Please see below (at your detailed comment on this issue) for more details.

I am less convinced by the "step-wise multiple linear regression." First, I don't understand the name – what is step-wise about it? Second, so many potential explanatory variables are used (which are said to be at least partly independent), that at least some of them are likely to have explanatory "power" by chance for the relatively short times series studied. I would be happier if the authors could clearly state a hypothesis, and then test it with a more limited set of explanatory variables consistent with the hypothesis. I am not a fan of a "throw everything at it and see what sticks" approach.

As also asked by the second reviewer, the step-wise multiple linear regression is explained in more detail in the paper. We indeed used a very large set of potential explanatory variables (ranging from 103 to 194, depending on the region), but the statistical test only retains on average 7 to 8 explanatory variables out of this list as contributing significantly to the correlation coefficient. We therefore believe in the explanatory power of this approach to select the most relevant variables for the IWV time series.

For information, we developed and applied the step-wise multiple linear regression initially to explain the time variability of the total ozone column at Uccle, Brussels. In this area, the relevant explanatory variables

(solar flux, QBO, EESC, stratospheric aerosols, EP flux) are well established by other studies and could be confirmed by our algorithm.

In this paper, we do not claim with our multiple linear regression approach that explanatory variable X can explain part of the IWV time variability of dataset X at station Z. Instead, we group the stations in regions (guided by the analyses of the frequency distribution and the seasonal variation) and discuss only the dominant explanatory variables for that specific region, for the three datasets used. By doing this, we are confident to cancel out to a certain extent the uncertainty related to the identification of specific explanatory variables for specific IWV time series.

From the literature, some obvious candidates for IWV explanatory variables could be determined, like the surface temperature, surface pressure, NAO, ENSO (see e.g. Wagner et al., 2006, Shi et al., 2018, Wang et al., 2018), Pacific Decadal Oscillation (PDO, see Shi et al., 2018). Rather than trying to confirm these links between IWV and those explanatory variables, we aim at finding out which are the most relevant variables by our empirical approach here. In this context, it should be noted as well that indices like ENSO have been defined rather arbitrarily (depending on the geographic location of the available weather stations) and are purely empirical as well. So there is already some arbitrariness in the whole matter. These last lines have also been added to the manuscript.

There are numerous cases of strange English usage/wording, some of which I mention below, but there are far more than I can explicitly call out. I recommend that the paper be edited by a native English speaker.
We went through the entire manuscript again, and try to avoid alternative English usage/wording than the usual formulations in scientific papers.

Some more detailed comments below:
Page 3, line 20-22. Strange Wording ("vastly"). Also, what does neutral mean in this context?
Neutral means here: nonionized component of the atmosphere (so, not the ionosphere). We explained this in the manuscript and changed "vastly" by "located mostly".

Page 3, line 30. Strange wording ("disposed of"). Maybe change to "This process results in a world-wide. . .."
Done.

Page 4, line 29. "downsized"? How was the conversion from 5-minute observations to 6 hourly observations performed? And why is this needed/appropriate considering that the reanalysis can be considered a snapshot at the synoptic times? (I'm not saying that the downsizing is wrong, I just want it explained better.
We changed the sentence into "We did not apply any time interpolation, so that, as the ERA-Interim reanalysis is only available at 0, 6, 12, and 18h UTC, the resulting GPS IWV dataset is also to the utmost available at these mentioned times."

Page 8. I can't follow the discussion of the clear sky/cloudy biases. Is the GOMESCIA only available in clear-sky conditions? If so, wouldn't it be expected that the GOMESCIA is biased low compared to measurements available in all-sky conditions?
On page 8, we wrote that "Looking at the biases, we found that 70% of the GPS stations have a negative IWV bias with respect to ERA-Interim, while 60% of the stations have a positive bias compared to GOMESCIA. Especially for sites in Europe, Southeast Canada and East USA, GOMESCIA shows a dry bias with respect to GPS, which in turn is dry-biased compared to ERA-Interim." So, indeed, GOMESCIA is biased low (dry bias) compared to both ERA-Interim and GPS. We tried to be clearer about this by linking "positive bias" to "wet bias" and "negative bias" to "dry bias" immediately.

Page 9 and 10. Discussion of different distribution classes at different locations. It is unclear how the sorting into classes was performed. Are all types of fits tried, and then some criterion applied? Please explain more clearly.
Indeed, we first fitted the different distribution classes separately (Gaussian, single lognormal, 2 lognormals) by means of a non-linear least squares fit to the frequency distributions of a station. A first, automated sorting into the different classes has been applied, based on the value of the chi-square goodness-of-fit statistic. Therefore, those values are compared for the Gaussian and single lognormal fit and the distribution with the lowest value is chosen. To sort between the different lognormal classes (single, shouldered, and bimodal), we defined some range intervals for the chi square of the standard lognormal fit (resp. $X^2 < 0.003$ for lognormal, between 0.003 and 0.01 for shouldered lognormal and above 0.01 for bimodal). These limiting values have been determined beforehand by comparing the chi square values and the quality of the fits (plots) by eye. This automated procedure is then used as a guidance (proposal) for the interactive procedure, in which we show all the frequency distribution fits of the different classes (and their chi squares) in one plot. In resp. 80% and 75% of the cases for respectively ERA-Interim and GPS, the

class proposed by the automated procedure have been adopted. The largest difference between the automated and interactive procedure took place for the Gaussian distribution, which was largely overestimated by the automated procedure. This procedure has now also been explained in more detail in the manuscript:

"In this work, we used a non-linear least squares fit to compute (separately) Gaussian, lognormal, and bimodal distribution functions for the IWV distributions observed at each GPS sites, making use of the same formula for the lognormal distribution function as in Foster et al. (2006). The sorting into the different categories was based for 75-80% of the sites on the values of the chi-square goodness-of-fit statistic (e.g. Gaussian if this statistic is lower than the lognormal one, a limiting value for the lognormal fit statistic to discriminate between a single and bimodal lognormal distribution) and determined interactively for the remaining sites by checking by eye the different frequency distribution fits. Examples of each category are given in Fig. 3a-b-d. We added an extra category, in between a lognormal and bimodal distribution, also in terms of the range of the chi-square goodness-of-fit statistic for the single lognormal distribution fit for these sites. For this category, there is one clear lognormal distribution which characterises the majority of the distribution, but an additional, secondary lognormal distribution (most often at the higher IWV side, an upper mode) is needed to explain the overall frequency distribution "satisfactorily", i.e.in terms of the chi-square goodness-of-fit statistic and/or by eye. We call it a "shouldered" lognormal distribution, see Fig. 3c."

Also, if the various distributions mostly occur because of seasonal cycles, maybe it would be good to show analysis after the seasonal cycle is removed??

We included an extra figure in our manuscript and extended/grouped this analysis: "To illustrate the impact of the seasonal variability on the shapes of the frequency distributions, we show in Fig. 5 the classification of the sites after the seasonal cycle has been removed from their time series (by subtracting the overall monthly means), both for GPS and ERA-Interim. A first thing to note is that no more bimodal distributions are present, but are replaced (mostly) by single lognormal or even Gaussian distributions. In particular, it is now clear that the dominance of the original bimodal distribution for the Asian sites (see Fig. 4a) is linked to the seasonal behaviour due to the monsoon, which is responsible for the reverse lognormal distribution with high median value, the strong upper mode, with the lower mode being caused by the dry season. The bimodality for the (sub)tropical sites is caused by the seasonal IWV variation too, and the reverse lognormal distribution is now very prominent for the deseasonalized IWV time series of (sub)tropical coastal or island sites (see Fig. 5). Another striking difference between the classification of the frequency distributions from the original and deseasonalized time series is situated in Europe: after removing the seasonal cycle, the dominating shouldered lognormal distributions in Fig. 4a are turned into standard lognormal distributions in Fig. 5a. So, in Europe, the shouldered lognormal distribution originates from the seasonal IWV variation. A remarkable feature for North America is the very similar geographical distinction in the continent in terms of the GSD values in Fig. 4b and the classes of frequency distributions of the deseasonalized IWV times series (Fig. 5b): whereas the sites in western part of North America (low GSD values) have standard lognormal distributions, the central and east North American stations (higher GSD values) are best fitted by shouldered lognormal distributions. For those latter stations, the IWV variability caused by weather or inter-annual variability seems to be more complex (multimodal) than if the seasonal variability is added."

Page 11, lines 19-20. This could be tested by subsetting the reanalysis data so that it too has gaps at the same time as the GPS data.

Thank you for this suggestion. We investigated this, and, as a matter of fact, the explained variabilities, the correlation coefficients, and the percentage of sites with a biannual cycle are now 64%, 0.776 and 68% respectively for ERA-Interim, and 55%, 0.710 and 60% for GOMESCIA. These numbers lie in the same range as for the GPS dataset, GOMESCIA being even lower. This has been adapted in the manuscript as follows:"These values are 81% and 0.895 for ERA-Interim respectively, and 71% and 0.831 for GOMESCIA. In this context, it should be noted that a slightly higher number of sites were found to have a statistically significant contribution from a biannual cycle for ERA-Interim (79%) and GOMESCIA (75%) than for GPS (69%), which also contribute to a better linear regression representation. The worse parameterisation by harmonics for the GPS dataset can be explained by the presence of gaps in the IWV monthly mean time series: if we subset the ERA-Interim and GOMESCIA monthly means to the GPS time series, the explained variabilities, correlation coefficients, and percentage of sites with a biannual cycle now lie in the same range as for the GPS dataset (even lower for GOMESCIA)."

Page 16. I wonder if the reason for the poor fitting for the west coast of North America sites it due to the fact that for much of this region, the rainy season is in the winter, where the temperatures are low. In the summer, it is dry but warmer, so there is not that much of a change in IPW. Other regions, such as eastern NA or Europe, there is still significant rainfall in the summer season, leading to a much larger correlation with surface temperature.

Thank you for this very valuable suggestion. Our data indeed shows the seasonal variations in surface temperature, IWV and precipitation for the majority of the sites of the west coast of North America. We included your suggestion in the manuscript, but we must confess that we have no idea how we might check it with our approach. Your suggestion is also consistent with the amplitudes of the seasonal cycle being

larger for the central and eastern sites in Northern America than for the western coast sites. So we also mentioned it in Sect. 5, where we included a discussion of the geographical pattern of the seasonal cycle, as requested by the first reviewer.

---

## Editor Comment (EC1) · Olivier Bock (Editor) · 7 Apr 2019

Editor comments on acp-2018-1170, revised manuscript (version 4), Interpreting the time variability of world-wide GPS and GOME/SCIAMACHY integrated water vapour retrievals, using reanalyses as auxiliary tools, by Roeland Van Malderen, Eric Pottiaux, Gintautas Stankunavicius, Steffen Beirle, Thomas Wagner, Hugues Brenot, and Carine Bruyninx

The word "tools" in the title of the manuscript is not very nice. I suggest replacing "…using reanalyses as auxiliary tools" by something like "by comparison with reanalyses".

P4L20-24: This sentence can be removed as it makes reference to other data sets which do not fit the scope of the study.

P4L28-29: "the 5-minute resolution IGS repro 1 ZTD estimates are downsized to a 6-h time resolution" downsized is not a proper term and does not explain how you converted the 5-min data to 6-hourly data. Please explained if either you selected the 5-min values that fall exactly on the 00, 06, 12, or 18 UTC times, or the nearest in time within some window, or if you averaged several 5-min values over some interval centred on the UTC times. Note that using the 5-min values instead of time-averaged values results in noisier time series with more gaps. Though this may not be a major issue when monthly means are computed afterwards, it should be mentioned should this be the way you handled the time sampling.

P5L7-12: why do you use surface data from ERA-Interim? They need to extrapolated from the model surface to the height of the GPS stations whereas pressure level data can be interpolated with higher accuracy to the height of the stations. Extrapolation of the model surface pressure to the altitudes of the GPS stations is known to introduce spurious seasonal signals and is not recommended (see my presentation at the GNSS4SWEC workshop in Reykjavik, 2016).

Bock O., A reference IWV dataset combining IGS repro1 and ERA-Interim reanalysis for the assessment of homogenization algorithms, 3rd COST ES1206 Workshop, 8-11.03.2016, Reykjavik, Iceland.

The motivation for including the NCEP/NCAR reanalysis 1 is not explained. This reanalysis has many known deficiencies in the representation of the water cycle and has been supplemented rapidly by NCEP/DOE reanalysis 2 and later by more modern global reanalyses (e.g. CFSR, MERRA, MERRA-2, JRA-55). Please justify the use of NCEP/NCAR reanalysis 1 or consider removing the results as they don't add much to the interpretation of time variability of the GPS IWV data.

P6L13-14: "The use of the NCEP/NCAR reanalysis is restricted to sensitivity analysis purposes because of its coarser spatial resolution." This statement should be revised since you actually report results using this reanalysis, namely at the end of Section 3 and in Section 4. But again, given the resolution limitations and known issues with this reanalysis I think these results are not relevant.

Here are the full references to our past studies comparing evaluating the NCEP/NCAR and NCEP/DOE reanalyses mentioned by one of the reviewers:

Bock, O., M.-N. Bouin, A. Walpersdorf, J.P. Lafore, S. Janicot, F. Guichard, A. Agusti-Panareda (2007), Comparison of ground-based GPS precipitable water vapour to independent observations and Numerical Weather Prediction model reanalyses over Africa. Q. J. R. Meteorol. Soc., 133, 2011-2027, DOI: 10.1002/qj.185

Bock, O., and M. Nuret (2009) Verification of NWP model analyses and radiosonde humidity data with GPS precipitable water vapor estimates during AMMA. Weather Forecast., 24: 1085-1101 DOI:10.1175/2009WAF2222239.1

Section 3: Two of the referees questioned about the time sampling and missing data issues and I could not find clear answers to these questions. When you compare the monthly mean IWV data from the three datasets and report statistical results (bias, standard deviation of differences, etc.) it is important to know how you computed the monthly means and how you handled the missing data. For example, did you put a limit on the minimum number of 6-hourly values entering into the monthly means? Did you compute the biases as the mean of differences or difference of means? (i.e. are the time series time-matched beforehand or not?).

Why did you choose to illustrate only the correlations (Fig. 2) in section 3? Please justify.

P8L15: you report a $R^2$ value of 0.975 here but in the Appendix the value in the first row is 0.962. Shouldn't it be the same?

P8L17-22: "surface pressure has a larger impact than other variables". Based on the materiel in the Appendix, provide a quantitative estimate of uncertainty for the IWV data that are used in this study.

P8L20-21: ERAI is not a "local" data but represents averages over the size of grid cells (0.75° x 0.75°)

Maybe a reference to Bock and Parracho, ACPD, 2019 (this Special Issue) can be useful here regarding the representativeness issues.

P8L26: "Looking at the biases" not easy since you don't show them. Please reformulated. Maybe provide results in a Table to support this discussion.

P9L3: "The standard deviations are smallest between GPS and ERA-interim": did you compare the standard deviations (variability)? Or do you (improperly) refer to the standard deviation of differences here?

P9L4: "Here, the impact between the different observations times (= satellite overpass times) at the sites for GOMESCIA compared to GPS and ERA-interim should be highlighted." Please explain the problem and evaluate the impact. A reference to Alraddawi et al., AMT, 2018 (this Special Issue) may be useful here.

P9L12: you should choose whether you think the results are "very similar" or "slightly worse". And I don't understand how the IWV comparison with NCEP can say something on the conversion using ERAI data? It may be that GPS IWV converted from ERAI data is more consistent with ERAI IWV data because the same reanalysis is used.

Section 4: it is not clear if you used monthly data or data with a higher time sampling.

P10L25: I think the NCEP results are not relevant because of the above-mentioned limitations. Moreover, since the results are only mentioned and not actually shown, I suggest to remove this sentence.

P17L27: I am not sure it is a good idea to use the NCEP results in the correlation analysis because of the above-mentioned limitations. Why don't you use the ERAI data instead?

Appendix: The presentation of Table A1 is not clear. I think it would be more clear to write out the compared data sources, e.g., for Tm and Ps in section [b] horizontally: Tm=ERA, Ps=ERA, and vertically: Tm=ERA, Ps=SYNOP; etc. Move the comparisons of Tm and Ps values from sections [b, c, d] to dedicated sections. For these comparisons the units are not mm. Section [a] is also a special case which should be specified as IWV=ERA; and IWV=GPS with Tm=ERA, Ps=ERA.

The text of the Appendix should be slightly revised. Strictly speaking, the sensitivity is quantified by the partial derivatives of IWV wrt Tm, Ts, and Ps, i.e. it is actually addressed only in the first paragraph (note that this paragraph should also be clarified and completed, especially quantify the impact quoted in the last sentence; maybe a reference to the PhD report of A. Parracho would be relevant here for further reading on this topic). The rest of the Appendix quantifies uncertainties (and not sensitivity) to the use of various datasets. This is useful but remains specific to the tested datasets. It might be interesting to add also the comparison between NCEP and ERA parameters (mentioned in the discussion but not reported in the Table).

Parracho, A. C., (2017) Study of trends and variability of atmospheric integrated water vapour with climate models and observations from global GNSS network, PhD report, Université Pierre et Marie Curie, Paris, France, http://www.theses.fr/2017PA066524